# Limiting oxidative DNA damage reduces microbe-induced colitis-associated colorectal cancer

Thergiory Irrazabal [1], Bhupesh K. Thakur[1], Mingsong Kang[1], Yann Malaise[1], Catherine Streutker[2], Erin O. Y. Wong[3], Julia Copeland[4], Robert Gryfe[5], David S. Guttman [4,6], William W. Navarre [3] & Alberto Martin [1✉]

Inflammatory bowel disease patients have a greatly increased risk of developing colitis-associated colon cancer (CAC); however, the basis for inflammation-induced genetic damage requisite for neoplasia is unclear. Using three models of CAC, we find that sustained inflammation triggers 8-oxoguanine DNA lesions. Strikingly, antioxidants or iNOS inhibitors reduce 8-oxoguanine and polyps in CAC models. Because the mismatch repair (MMR) system repairs 8-oxoguanine and is frequently defective in colorectal cancer (CRC), we test whether 8-oxoguanine mediates oncogenesis in a Lynch syndrome (MMR-deficient) model. We show that microbiota generates an accumulation of 8-oxoguanine lesions in MMR-deficient colons. Accordingly, we find that 8-oxoguanine is elevated in neoplastic tissue of Lynch syndrome patients compared to matched untransformed tissue or non-Lynch syndrome neoplastic tissue. While antioxidants reduce 8-oxoguanine, they do not reduce CRC in Lynch syndrome models. Hence, microbe-induced oxidative/nitrosative DNA damage play causative roles in inflammatory CRC models, but not in Lynch syndrome models.

[1] Department of Immunology, University of Toronto, Toronto, ON M5S 1A8, Canada. [2] Department of Laboratory Medicine, St. Michael's Hospital, Toronto, ON M5B 1W8, Canada. [3] Department of Molecular Genetics, University of Toronto, Toronto, ON M5S 1A8, Canada. [4] Centre for the Analysis of Genome Evolution & Function, University of Toronto, Toronto, ON M5S 3B2, Canada. [5] Department of Surgery, Mount Sinai Hospital, Toronto, Ontario M5G 1X5, Canada. [6] Department of Cell & Systems Biology, University of Toronto, Toronto, ON M5S 3G5, Canada. ✉email: alberto.martin@utoronto.ca

nflammatory bowel diseases (IBD), such as Crohn's disease and ulcerative colitis, are positively associated with colitis-associated colon cancer (CAC)[1]. Indeed, after hereditary syndromes, such as Familial adenomatous polyposis and Lynch syndrome, chronic inflammation in the colon is considered one of the largest identified risk factors for the development of colorectal cancer (CRC). While cancer is known to be a genetic disease, the sources of genetic damage caused by inflammation in CAC are not known.

Factors that might directly mediate genetic damage in the inflammatory milieu include reactive oxygen species (ROS) and reactive nitrogen intermediates (RNI) produced by activated immune cells. During inflammation copious amounts of superoxide ($O_2^{\bullet-}$), and the highly diffusible and membrane permeant nitric oxide radical ($^\bullet$NO) are generated by the NADPH oxidase enzyme and the inducible nitric oxide synthase (iNOS), respectively[2,3]. However, DNA-damaging ROS can also come from endogenous mitochondrial respiration, cellular oxidases, and peroxidases[2]. Superoxide, which does not directly damage DNA, can dismutate to hydrogen peroxide ($H_2O_2$), which can diffuse and react with free transition metals via Fenton chemistry to produce the highly reactive hydroxyl radical[2]. $^\bullet$NO itself is poorly reactive with DNA, however, it can combine with $O_2^{\bullet-}$ to generate the highly reactive congener peroxynitrite ($ONOO^-$)[4]. These products can damage DNA via strand breakage or through the oxidation of guanine, the nucleotide with the highest oxidation potential[4]. Guanine oxidation to generate 8-hydroxy-2′-deoxyguanosine or 8-oxo-7,8-dihydro-2′-deoxyguanosine (8-oxoG) is mutagenic. ROS produces the oxidative DNA lesion 8-oxoG, while nitric oxide produces various forms of DNA lesions including 8-oxoG (refs. [4,5]).

Cellular systems that repair oxidative lesions in DNA include the base-excision repair (BER), carried out by 8-oxoguanine DNA glycosylase (OGG1) and MutY DNA glycosylase (MUTYH), and mismatch repair (MMR)[6]. ROS leads to guanine oxidation forming C:8-oxoG pairs, which are substrate for OGG1-initiated BER. During replication, an adenine can be inserted opposite 8-oxoG, generating an A:8-oxoG pair that are processed through MUTYH-initiated BER. The MMR pathway can recognize and repair these mismatches as well, but if this mismatch is left unrepaired, a second round of replication will generate an A:T pair, giving rise to a C:G > A:T transversion mutation. The primary role of the MMR system is to repair DNA lesions that occur as a result of replication errors and specific types of mismatches, including A:8-oxoG base pairs[6–8]. Notably, MMR defects are strongly associated with CRC. Lynch syndrome patients harboring mutations in MMR genes, such as *MSH2*, *MSH6*, *PMS2*, or *MLH1*, represent ~5% of all CRC cases[9]. Furthermore, an additional 15% of sporadic CRC tumors harbor mutations or have epigenetically silenced MMR genes.

Some studies have focused on the role that bacterial toxins might play in directly damaging host DNA. However, CAC might also be a more general consequence of immune-generated ROS and RNI produced during intestinal inflammation. The resulting DNA damage, under normal circumstances, might be repaired by the MMR system. Cells lacking MMR could be exceptionally susceptible to oxidative damage, and lack an appropriate apoptotic response, and hence would further accumulate the requisite mutations to drive neoplasia. A testable prediction from this model is that the therapeutic reduction of ROS- or RNI-induced genetic lesions would be sufficient to prevent cancer initiation in the inflamed colon. Some clinical trials suggest that neutralizing ROS with antioxidants might reduce CRC (refs. [10,11]), although not all studies are in agreement[12,13]. However, only one of these studies stratified patients based on patient or tumor genetics[11], and none have assessed whether antioxidants impact oncogenesis

in IBD patients. Importantly, one clinical trial found that vitamin C (VitC) reduces oxidative stress in IBD patients[14], and hence might be effective at neutralizing DNA damage caused by ROS and reducing CAC.

Colorectal inflammation and the production of intestinal ROS and RNI are influenced most strongly by pathogens and gut-associated microbes. Colibactin-producing *Escherichia coli* (i.e., *E. coli* NC101), enterotoxigenic *Bacteroides fragilis* (ETBF), and *Helicobacter hepaticus* (*H. hepaticus*) cause tumorigenesis in mice by stimulating a chronic inflammatory response[15–18]. In this report, we examine the interplay between microbiota, genetics, and inflammation, to dissect the mechanism by which gut microbiota induce CRC. Using IL10$^{-/-}$ mice, we find that infection with *E. coli* NC101, ETBF, or *Helicobacter species*, or tissue damage caused by dextran sodium sulfate (DSS) leads to inflammation, which generates an accumulation of oxidative DNA damage in colonic tissue. Because the MMR system repairs 8-oxoG, we test whether oxidative DNA damage is a mediator of carcinogenesis in a Lynch syndrome colon cancer model. We show that gut microbiota, partially through the production of butyrate, induces ROS and the accumulation of 8-oxoG lesions and double-strand DNA breaks in MMR-deficient cells. Notably in the inflammatory mouse models, DNA damage and tumorigenesis are reduced by treating with the antioxidants VitC or N-acetyl cysteine (NAC), or a compound that inhibits the production of RNI. On the other hand, in Lynch syndrome models, VitC and NAC treatment reduce oxidative DNA damage but not tumorigenesis. Our results suggest that IBD patients might benefit from the use of antioxidants or inhibitors of iNOS to blunt the oxidative DNA damage that can precipitate CRC.

## Results

**Helicobacters induce inflammation, dysbiosis, and polyposis.** We utilized different murine models of CAC; two bacterial infection models, and an abiotic DSS treatment, to assess whether the oncogenic mechanisms are similar in each. In the first model, we infected IL10$^{-/-}$ mice with enterohepatic *Helicobacter* species, some of which have been shown to induce CRC[19]. IL10$^{-/-}$ mice infected with two *Helicobacter* species (IL10$^{-/-}$I), namely a combination of *Helicobacter typhlonius* (*H. typhlonius*) and *Helicobacter mastomyrinus* (*H. mastomyrinus*; Supplementary Fig. 1a), developed colitis (Fig. 1a, b) and colonic tumors (Fig. 1c, Supplementary Fig. 1b). Littermate control IL10$^{+/-}$ or uninfected IL10$^{-/-}$ mice did not developed major signs of colitis or tumorigenesis (Fig. 1a–c). In separate experiments, we also gavaged 4-week-old IL10$^{-/-}$ and IL10$^{+/-}$ mice with *H. hepaticus* (Supplementary Fig. 1c). As with the other *Helicobacter* infections, *H. hepaticus* infection triggered colitis and tumorigenesis in IL10$^{-/-}$ mice (Fig. 1d–f, Supplementary Fig. 1d). On the other hand, infection of 4-week-old IL10$^{-/-}$ mice with *Citrobacter rodentium* (*C. rodentium*) (Supplementary Fig. 1e) did not lead to colitis or polyp formation in the colon (Fig. 1d, f). Colitis development was associated with a gradual shift in the composition of the fecal bacterial microbiota (i.e., 'dysbiosis') in *H. hepaticus*-infected IL10$^{-/-}$ mice (Fig. 1g, Supplementary Fig. 1f). Littermate control IL10$^{+/-}$ mice, which did not develop colitis or colon polyps after infection (Fig. 1d–f), also did not develop dysbiosis (Fig. 1h, Supplementary Fig. 1f). Collectively, these data show that enterohepatic *Helicobacter* species can trigger sustained inflammation and dysbiosis in IL10$^{-/-}$ mice that leads to the development of colon tumors. Given that heterozygous mice did not develop either colitis or neoplasia, these findings support the model that a single pathogen + susceptibility gene = colitis[20] and, ultimately, tumorigenesis.

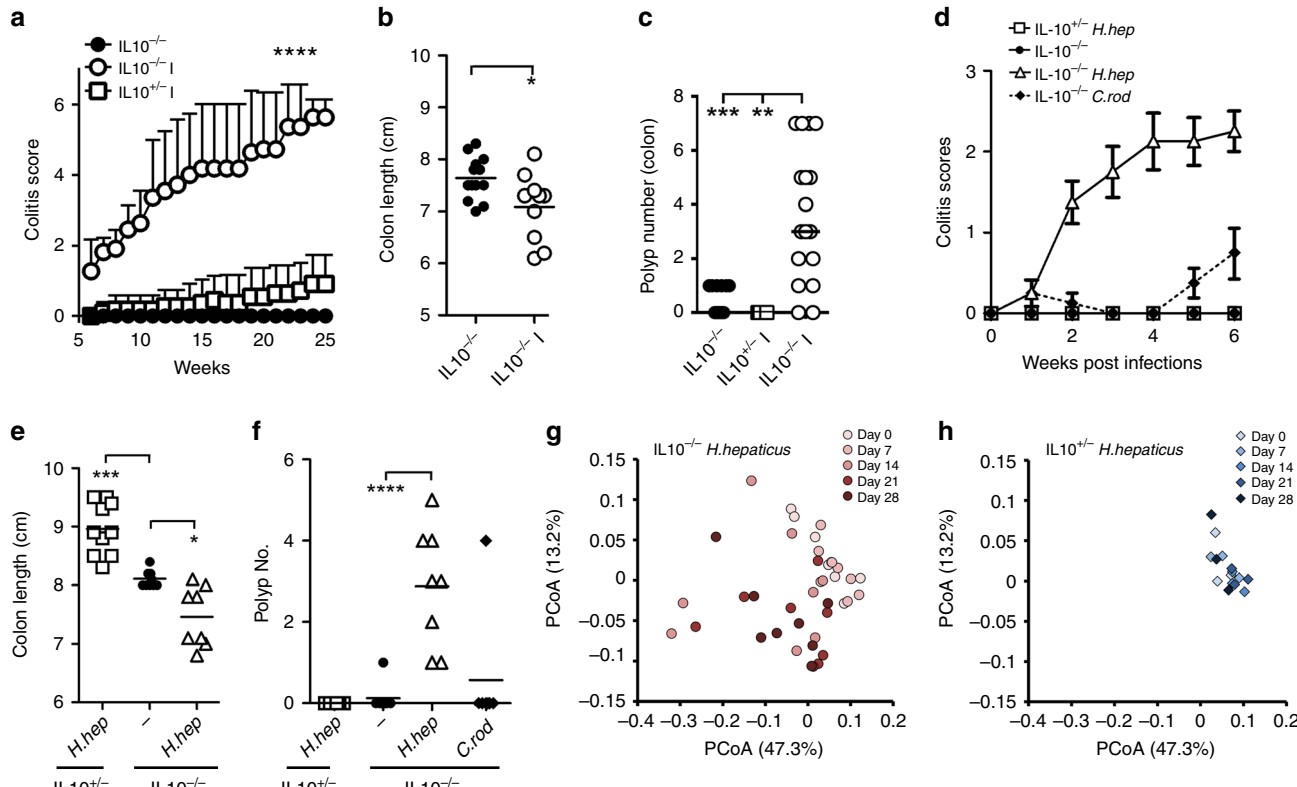

**Fig. 1 *Helicobacter* species induce inflammation, dysbiosis, and colon tumorigenesis in IL10$^{-/-}$ mice. a** Colitis scores for IL10$^{-/-}$ mice compared to IL10$^{-/-}$ (IL10$^{-/-}$I) and IL10$^{+/-}$ (IL10$^{+/-}$I) infected with two *Helicobacter* species: *H. typhlonius* and *H. mastomyrinus*. To account for disease, each of the following parameters was given a value of either 0 or 1: stool with excreted mucus, rectal inflammation, rectal prolapse, bloody stool, over 5% weight lost, and moribund, leading to a maximum grade of 6. $N = 33$ mice were examined. Data are presented as mean values + SEM. Two-way ANOVA $p < 0.0001$. **b** Colon length measured in 9-week-old *Helicobacter*-infected or -uninfected IL10$^{-/-}$ mice. Shortening of the colon is indicative of colitis. $N = 30$ mice were examined. **c** Polyp number was analyzed in the colon of 9-week-old *Helicobacter*-infected (IL10$^{-/-}$ and IL10$^{+/-}$ littermates) or -uninfected IL10$^{-/-}$ mice. $N = 35$ mice were examined. **d** 3- or 4-week-old IL10$^{-/-}$ or IL10$^{+/-}$ mice were inoculated by oral gavage with *H. hepaticus* or *C. rodentium*, and colitis was monitored for 6 weeks. $N = 34$ mice were examined. Data are presented as mean values ± SEM. **e** Colon length measured on 10-week-old mice from the indicated genotypes and treatments. $N = 24$ mice were examined. **f** Polyp number was analyzed in the colon of 10-week-old mice from the indicated genotypes and treatments. $N = 30$ mice were examined. **g** Principal coordinate analysis of 16 S rRNA gene-sequencing analysis of fecal bacteria obtained from IL10$^{-/-}$ mice infected with *H. hepaticus*. Fecal samples were obtained before (day 0) and after *H. hepaticus* infection, for four consecutive weeks. Each dot represents one mouse. $N = 8$ mice were examined. **h** Same as **g**, except that IL10$^{+/-}$ mice were used. $N = 3$ mice were examined. Data in **b**, **c**, **e** and **f** were analyzed using the two-sided non-parametric *t*-test Mann–Whitney; *$p < 0.05$, **$p < 0.01$, ***$p < 0.001$, ****$p < 0.0001$.

**Antioxidants reduce polyps in *Helicobacter*-infected mice.** To test whether ROS or RNI contributed to genetic damage, IL10$^{-/-}$ mice infected with *H. typhlonius* and *H. mastomyrinus* were given the iNOS inhibitor L-N$^6$-(1-Iminoethyl) lysine dihydrochloride (L-NIL) or the antioxidant NAC in their drinking water for 8 weeks post infection. Notably, neither treatment affected various metrics of inflammation, including colon length, and neutrophil and lymphocyte mucosal infiltration (Fig. 2a, b). Strikingly, however, L-NIL and NAC reduced polyposis in mice infected with a combination of *H. typhlonius* and *H. mastomyrinus* (Fig. 2c, d). Antibody staining revealed that levels of 8-oxoG in the nuclei of colon epithelial cells of IL10$^{-/-}$ I mice was markedly reduced by L-NIL or NAC treatment (Fig. 2e, Supplementary Fig. 3a, b). Using mice singly infected with *H. typhlonius* or *H. mastomyrinus*, we showed that VitC treatment also reduced polyposis (Fig. 2c, d) and levels of 8-oxoG in IL10$^{-/-}$ mice (Supplementary Fig. 3c–e) without major effects on inflammation or *Helicobacter* colonization (Fig. 2a, Supplementary Fig. 2). Together, these data suggest that oxidative DNA damage induced by ROS and RNI play central roles in polyp induction caused by *Helicobacter* species in IL10$^{-/-}$ mice.

**Antioxidants reduce polyps in DSS-treated IL10$^{-/-}$ mice.** To determine if oxidative DNA damage is generated in response to abiotic sources of inflammation, we treated IL10$^{-/-}$ and IL10$^{+/-}$ mice with 1% DSS in the drinking water for 1 week followed by 5 weeks of regular drinking water. After this point, inflammation, dysbiosis, and polyps were measured. While DSS treatment initially triggered colitis in both IL10$^{-/-}$ and IL10$^{+/-}$ mice, colitis scores returned back to baseline in control IL10$^{+/-}$ mice, whereas they remained high in IL10$^{-/-}$ mice until the end of the 6-week experiment (Fig. 3a, b, Supplementary Fig. 4a). DSS treatment induced a transient shift in the microbial composition both in IL10$^{+/-}$ and IL10$^{-/-}$ mice that mirrored the inflammatory scores (Fig. 3c, d, Supplementary Fig. 4c). However, colonic polyps were only observed in DSS-treated IL10$^{-/-}$ mice, and not in the DSS-treated IL10$^{+/-}$ mice (Fig. 3e, Supplementary Fig. 4b).

To test whether the oncogenic mechanism in the DSS IL10$^{-/-}$ model is similar to the *Helicobacter*-infected IL10$^{-/-}$ mice, we treated IL10$^{-/-}$ mice with 1% DSS for 1 week, then treated mice with L-NIL, NAC, or VitC for 4 weeks. L-NIL or antioxidants did not affect inflammation (Fig. 4a–c); however, NAC, VitC, and to

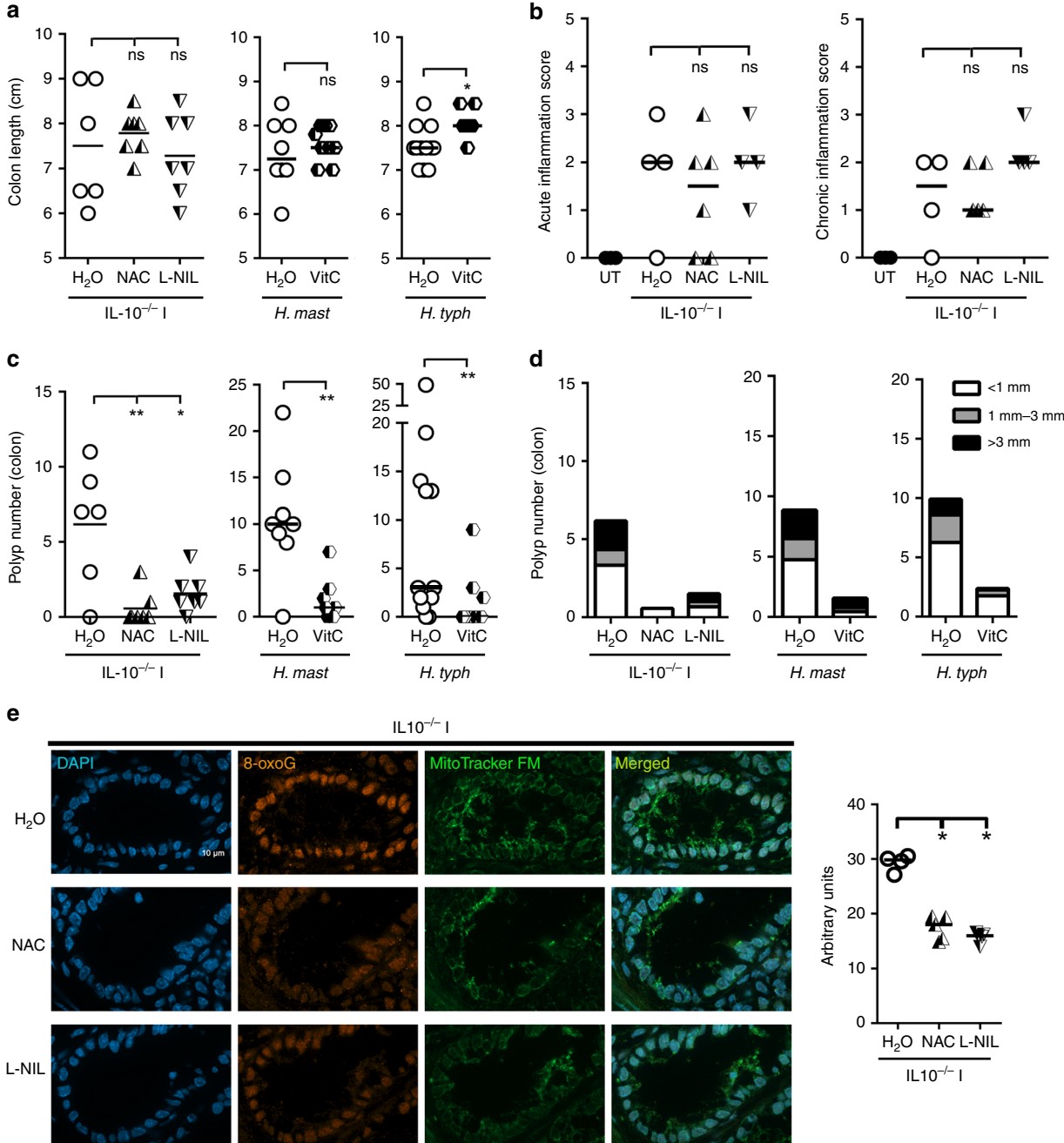

**Fig. 2 Antioxidants reduce *Helicobacter*-induced 8-oxoG and colon tumorigenesis. a** Colon length measured in 12-week-old *Helicobacter*-infected IL10$^{-/-}$ mice untreated or treated with L-NIL, NAC, or VitC. IL10$^{-/-}$ I indicates that mice were infected with both *H. mastomyrinus* and *H. typhlonius*. $N = 57$ mice were examined. **b** Acute and chronic inflammation were assessed histopathologically in *Helicobacter*-infected or uninfected IL10$^{-/-}$ mice. $N = 17$ mice were examined. **c** Polyp number was analyzed in the colon of 12-week-old *Helicobacter*-infected IL10$^{-/-}$ mice untreated or treated with L-NIL, NAC, or VitC. $N = 57$ mice were examined. **d** Polyps counted in **c** were graded according their size. **e** Immunofluorescence for 8-oxoG and MitoTracker in colon from *Helicobacter*-infected IL10$^{-/-}$ mice administered with L-NIL or NAC. Representative images of four independent experiments involving at least one mouse per group are depicted. Magnification 100×. Scale bar = 10 μm. Right panel: one dot represents the median intensity of fluorescence of 40 nuclei per mouse. Data in **a**, **b**, **c** and **e** were analyzed using the two-sided non-parametric *t*-test Mann–Whitney; *$p < 0.05$, **$p < 0.01$; ns nonsignificant.

lesser extent, L-NIL reduced colon tumors and 8-oxoG in colon epithelial cell nuclei (Fig. 4d, e, Supplementary Fig. 4d, e). Hence, oxidative DNA damage is one of the main mechanisms for neoplasia in an abiotic model of CAC.

**Antioxidants reduce polyps in *E.coli* NC101$^+$/ETBF$^+$ mice.** To investigate whether these findings can be applied to another CAC-infection model, we colonized IL10$^{-/-}$ mice with a mix of

two bacterial strains that are enriched in tumors of patients with familial adenomatous polyposis[18]. Four-week-old IL10$^{-/-}$ mice were co-colonized with *E. coli* NC101, which expresses colibactin, and ETBF, which expresses the *B. fragilis* toxin, and were treated or untreated with L-NIL or VitC for 8 weeks. Sustained co-colonization was confirmed by PCR and was not affected by L-NIL or VitC treatments (Supplementary Fig. 5a). Colonized mice displayed increased expression of pro-inflammatory

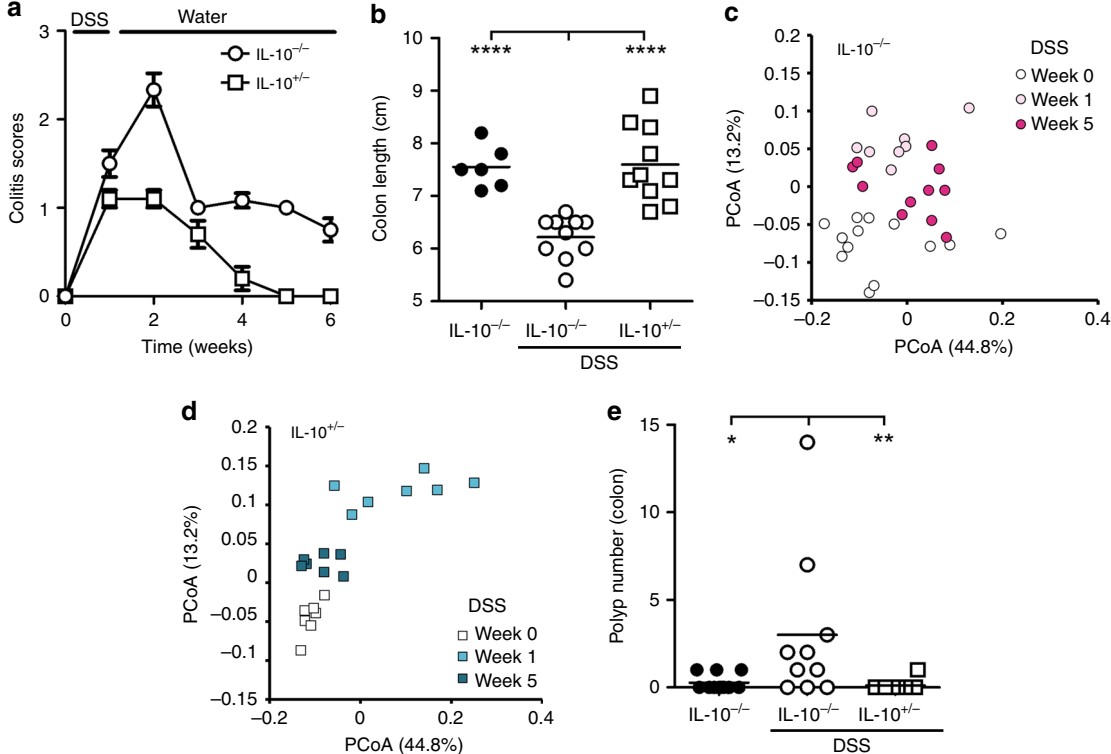

**Fig. 3 DSS-treated IL10$^{-/-}$ mice develop colon polyposis and dysbiosis. a** Four-week-old IL10$^{+/-}$ and IL10$^{-/-}$ mice were treated with 1% DSS for 1 week followed by 5 weeks of regular drinking water, and colitis development was monitored as in Fig. 1a. $N = 20$ mice were examined. Data are presented as mean values ± SEM. **b** Colon length was measured in 9-week-old mice of the indicated genotypes and treatments. $N = 26$ mice were examined. **c** Weighted UniFrac principal coordinate analysis of 16 S rRNA gene-sequencing analysis of fecal bacteria obtained from DSS-treated IL10$^{-/-}$ mice. Fecal samples were obtained before (day 0), 1, and 5 weeks after DSS treatment. $N = 13$ mice were examined. **d** Same as **c**, except that IL10$^{+/-}$ mice were used. $N = 7$ mice were examined. **e** Polyp number was analyzed in the colon of 9-week-old mice of the indicated genotypes and treatments. $N = 26$ mice were examined. Data in **b** and **e** were analyzed using the two-sided non-parametric $t$-test Mann–Whitney; *$p < 0.05$, **$p < 0.01$, ****$p < 0.0001$.

cytokines in the gut (Fig. 5a). However, colonization did not induce changes in colon length or alteration in inflammatory cell infiltration (Fig. 5b, c), although the cecums of colonized mice were decreased in size compared to uncolonized mice, which is indicative of mild inflammation (Fig. 5d). Strikingly, *E. coli* NC101 + ETBF-induced tumorigenesis in IL10$^{-/-}$ mice was reduced by L-NIL or VitC treatment (Fig. 5e), without affecting inflammation (Fig. 5a–d, Supplementary Fig. 5b). VitC and L-NIL also reduced infection-induced oxidative DNA damage (Fig. 5f, Supplementary Fig. 5c). Hence, scavenging ROS or inhibiting iNOS reduces oxidative DNA damage, preventing polyposis in spite of inflammation in three different CAC models. These data support the notion that ROS and RNI are primary sources of the genetic damage that leads to the tumor formation in the colon, regardless of the initial inflammatory insult.

**The MMR is the dominant repair pathway of 8-oxoG in colon**. DNA damage that occurs during normal metabolism or inflammation in heathy tissue is presumably repaired by systems like BER and MMR. MMR is defective in both familial (Lynch syndrome) and sporadic forms of CRC. To explore whether the role of MMR in preventing CRC relates to its ability to repair oxidative DNA damage, we employed murine models of Lynch syndrome. Our earlier study found that resident gut microbes could dramatically enhance CRC progression in Lynch syndrome mice, but left unanswered the direct underlying mutagenic mechanisms[21]. In this study, we did not induce inflammation but instead examined how endogenously produced ROS could impact progression of neoplasia. Notably, butyrate produced from

microbes is a major carbon source for colonic epithelial cell metabolism and can induce mitochondrial ROS production in intestinal epithelial cells cultured in vitro[22,23]. Hence, we tested whether ROS generated as a by-product of butyrate metabolism contributes to DNA damage (specifically the 8-oxoG lesion) and CRC in Lynch syndrome models.

To confirm that butyrate can induce ROS in the intestine, we cultured colon epithelial cells from *Msh2*$^{+/-}$ and *Msh2*$^{-/-}$ mice for 30 min with a physiological range of butyrate (0.5–2 mM). Cells were then treated with either DCFDA, which fluoresces when it reacts with intracellular $H_2O_2$, or Amplex Red, which fluoresces when it reacts with extracellular $H_2O_2$. The results indicate that *Msh2*$^{-/-}$ colon epithelial cells generate higher basal levels of ROS than *Msh2*$^{+/-}$ colon epithelial cells, and that butyrate treatment increased ROS production in both genotypes (Supplementary Fig. 6a, b). As expected, treatment with the glutathione precursor NAC reduced the amount of intracellular $H_2O_2$ more than extracellular $H_2O_2$ (Supplementary Fig. 6a, b).

To test whether butyrate is inducing ROS by acting as an inhibitor of histone deacetylases (iHDAC) or through its oxidative metabolism, we extracted colon epithelial crypts from mice treated with an antibiotic cocktail (ampicillin, metronidazole, and neomycin) for 3 weeks after weaning, which leads to a ~10$^3$-fold reduction in gut microbiota and a lack of butyrate in the colon[21]. We then treated these crypts with butyrate, palmitate (another fatty acid that is metabolized by β-oxidation), and trichostatin A (TSA; an iHDAC). We found that butyrate and palmitate induced mitochondrial ROS production in both

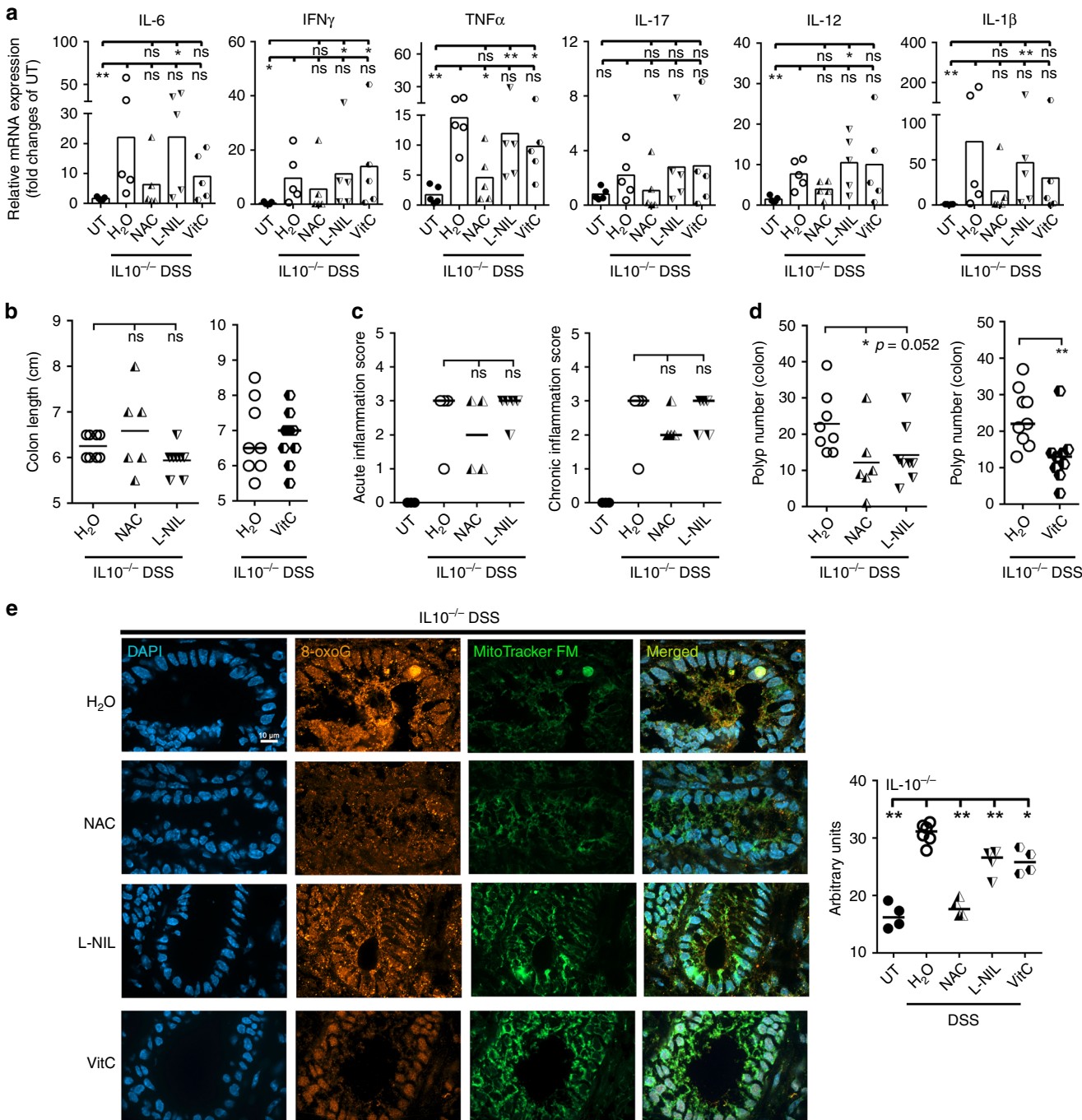

**Fig. 4 Antioxidants reduce DSS-induced oxidative DNA damage and polyposis in IL10$^{-/-}$ mice. a** Four-week-old IL10$^{-/-}$ mice were treated with 1% DSS for 1 week and then administered with water, L-NIL, NAC, or VitC for 4 weeks. cDNA levels of indicated inflammatory cytokines were quantified by qPCR. Relative mRNA expression was normalized to 1 for untreated IL10$^{-/-}$ mice. $N = 20$ mice were examined. **b** Colon length was analyzed in 9-week-old mice. $N = 39$ mice were examined. **c** Acute and chronic inflammation were assessed histopathologically in the colon of 9-week-old IL10$^{-/-}$ mice with the indicated treatments. $N = 16$ mice were examined. **d** Polyp number was analyzed in the colon of 9-week-old mice. $N = 40$ mice were examined. **e** Immunofluorescence for 8-oxoG and MitoTracker in colon from DSS-treated IL10$^{-/-}$ mice administered with antioxidants. Representative images of four independent experiments involving at least one mouse per group are depicted. Magnification 100×. Scale bar = 10 μm. Right panel: quantification of 8-oxoG immunofluorescence. One dot represents the median intensity of fluorescence of 40 nuclei per mouse as shown in Supplementary Fig. 4e. All data were analyzed using the two-sided non-parametric t-test Mann–Whitney; *$p < 0.05$, **$p < 0.01$; ns nonsignificant.

genotypes, while TSA did not induce an increase in ROS (Supplementary Fig. 6c). These data suggest that mitochondrial oxidative metabolism of butyrate is the main mechanism by which butyrate increases ROS in colon epithelial cells.

Since MMR repairs ROS-induced 8-oxoG lesions, we next tested whether 8-oxoG levels were elevated in colon epithelial cells from $Msh2^{-/-}$ mice with an intact microbiota. Indeed, the levels of 8-oxoG were increased in colon epithelial cells from $Msh2^{-/-}$ mice compared to controls (Fig. 6a, Supplementary Fig. 7a). This lesion was dependent on the microbiota since mice treated with an antibiotic cocktail for 3 weeks after weaning, were devoid of this genetic lesion (Fig. 6a, Supplementary Fig. 7a).

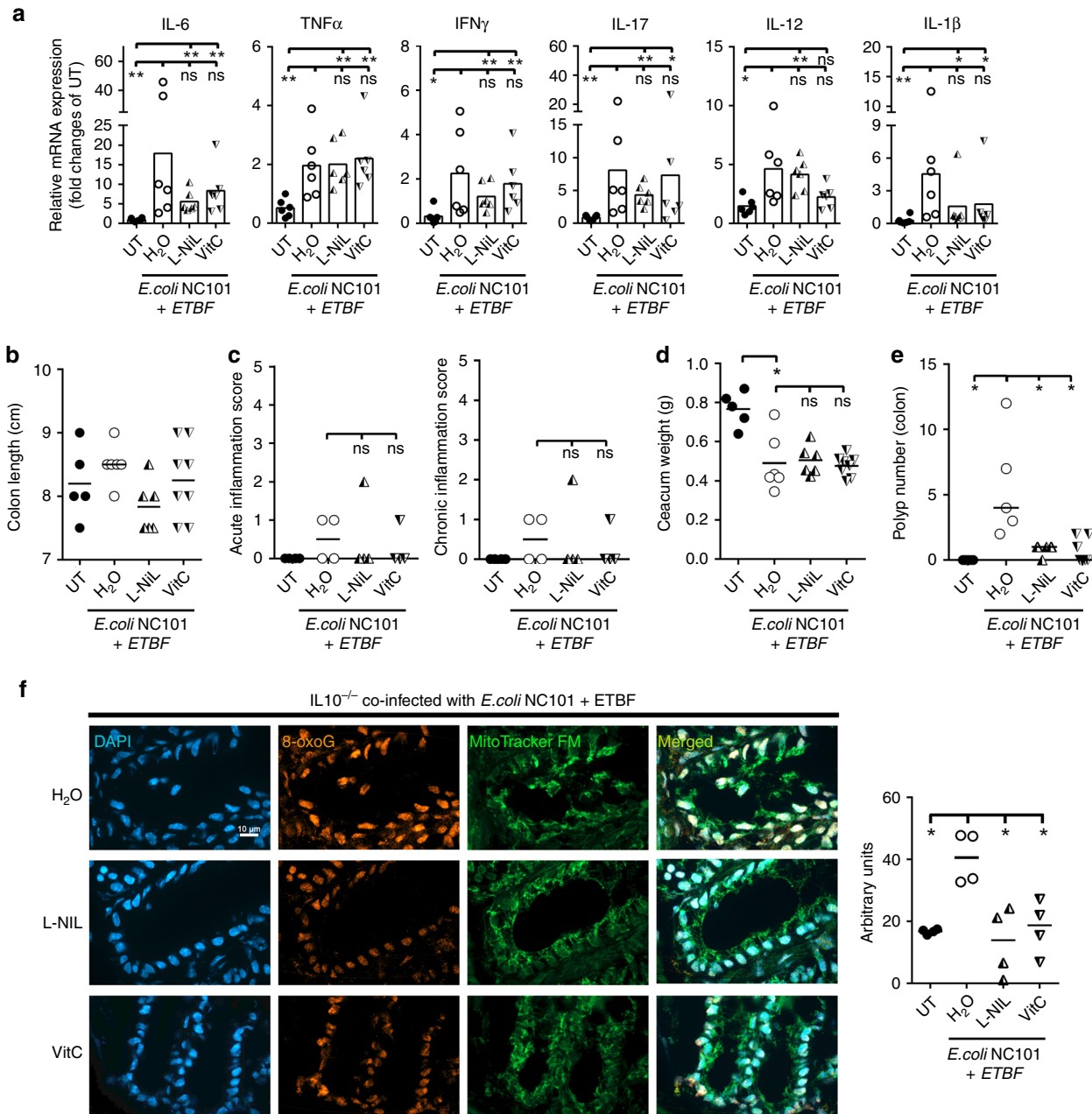

**Fig. 5 Antioxidants reduce *E. coli* NC101 + ETBF-induced polyposis in IL10 $^{-/-}$ mice. a** Four-week-old IL10$^{-/-}$ mice were inoculated by oral gavage with *E. coli* NC101 and ETBF, and treated or untreated with L-NIL or VitC for 8 weeks. cDNA levels of indicated inflammatory cytokines were quantified by qPCR. Relative mRNA expression was normalized to 1 for untreated IL10$^{-/-}$ mice. $N = 24$ mice were examined. Colon length **b**, acute inflammation and chronic inflammation **c**, cecum weight **d**, and colonic polyp number **e** were measured in 12-week-old IL10$^{-/-}$ mice of the indicated treatments. $N = 25$ mice were examined. **f** Immunofluorescence for 8-oxoG and MitoTracker in colon from *E. coli* NC101 and ETBF-infected IL10$^{-/-}$ mice administered with L-NIL or VitC. Magnification 100×. Scale bar = 10 μm. Right panel: quantification of 8-oxoG immunofluorescence shown to the left. One dot represents the median intensity of fluorescence of 40 nuclei per mouse as shown in Supplementary Fig. 5c. $N = 16$ samples were examined over three independent experiments. All data were analyzed using the two-sided non-parametric *t*-test Mann–Whitney; *$p < 0.05$, **$p < 0.01$; ns nonsignificant.

To test whether butyrate alone can produce 8-oxoG in mice, *Msh2*$^{+/-}$ and *Msh2*$^{-/-}$ mice were treated with the antibiotic cocktail for 3 weeks after weaning (week 3–6 post birth), and then rectally instilled with 100 μL of PBS or 0.5 mM butyrate for three consecutive days. Antibiotic-treated *Msh2*$^{-/-}$ mice rectally instilled with butyrate showed increased 8-oxoG compared to antibiotic/PBS-treated *Msh2*$^{-/-}$ mice or antibiotic/butyrate-treated *Msh2*$^{+/-}$ mice (Fig. 6a, Supplementary Fig. 7a), although the levels of 8-oxoG never reached those observed in untreated Msh2$^{-/-}$ mice. In addition, butyrate enemas induced a dose-

dependent increase in the double-strand DNA break marker γH2AX in the colon of *Msh2*$^{-/-}$ mice (Supplementary Fig. 7b). Taken together, these results suggest that microbial-produced butyrate is at least one source of ROS generation that lead to the accumulation of oxidative DNA damage in *Msh2*$^{-/-}$ colon epithelial cells.

To assess whether oxidative DNA damage is also elevated in tumors from Lynch syndrome patients (Table 1), we stained neoplastic tissue and flanking normal tissue with the 8-oxoG antibody. Tumors from Lynch syndrome patients typically

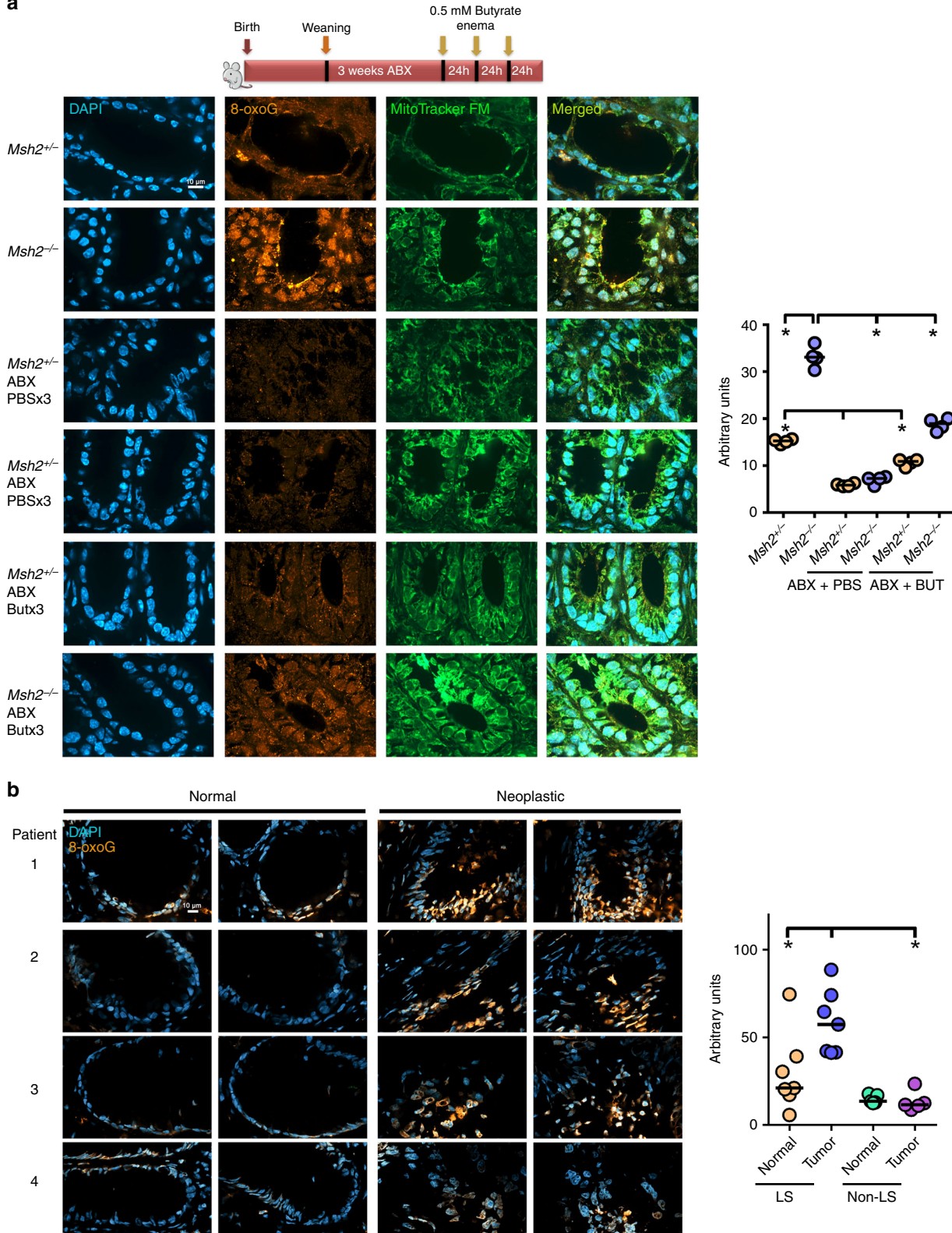

inactivate the unmutated MMR allele[24,25], and hence, tumors are expected to be MMR deficient, but not the flanking tissue. We found that 8-oxoG was elevated in neoplastic tissue compared to untransformed tissue from the same patient (Fig. 6b, Supplementary Fig. 8a). In addition, we examined tumor tissue from non-Lynch patients (Table 2) and find that the levels of 8-oxoG are normal in these tissues (Fig. 6b right panel, Supplementary

Fig. 8a, b). Hence, oxidative DNA damage accumulates to higher levels in murine and human tissues that are deficient in MMR indicating that the MMR system is the major DNA repair pathway for 8-oxoG in the gut.

**Antioxidants reduce 8-oxoG but not polyps in MMR mutants.** Since antioxidants reduced the levels of 8-oxoG in colon epithelial

**Fig. 6 The MMR is the dominant repair pathway of the 8-oxoG lesion in mouse and human colon. a** Top panel: schematic outlining the experimental interventions before immunofluorescence against 8-oxoG. Antibiotic cocktail was administered in drinking water for 3 weeks. Butyrate or PBS were administered for three consecutive days. Left panel: immunofluorescence for DAPI, 8-oxoG, and MitoTracker in colon from mice of the indicated treatments and genotypes. Magnification 100×. Scale bar = 10 μm. Right panel: quantification of 8-oxoG immunofluorescence. One dot represents the median intensity of fluorescence of 40 nuclei per mouse as shown in Supplementary Fig. 7a. $N = 24$ samples were examined over four independent experiments. **b** Left panel: immunofluorescence for DAPI and 8-oxoG in colon from four Lynch syndrome patients. Magnification 63×. Scale bar = 10 μm. Right panel: quantification of 8-oxoG immunofluorescence are shown for the Lynch patients (**b**, left panel) and non-Lynch patients (Supplementary Fig. 8b). One dot represents the median intensity of fluorescence of 40 nuclei per patient as shown in Supplementary Fig. 8a. $N = 12$ biopsies (plus 12 matched normal tissue) were examined over two independent experiments. All data were analyzed using the two-sided non-parametric $t$-test Mann–Whitney; $^*p < 0.05$.

**Table 1 Lynch syndrome patient information and tumor characteristics.**

| Patient | Gene | Mutation | Cancer stage | Cancer site | Age at diagnosis | Gender | Other cancer[a] |
|---|---|---|---|---|---|---|---|
| 1 | MSH2 | Frameshift | T3, N0, M0 | Hepatic flexure | 36.8 | Female | Endometrial cancer @40.5 |
| 2 | PMS2 | Missense | T4a, N0, M0 | Transverse | 54.0 | Male | Prostate cancer @60.4 |
| 3 | PMS2 | Frameshift | T4b, N0, M0 | Rectum | 29.2 | Male | Testis cancer @36.4 |
| 4 | PMS2 | Frameshift | T3, N0, M0 | Sigmoid | 64.9 | Male | Hepatic flexure cancer @71.2 |
| 5 | MSH2 | Deletion | T4, N0, M1 | Cecum | 60 | Female | Endometrium @33 |
| 6 | MSH2 | Deletion | T4, N2, M0 | Rectum | 43 | Female | Transverse colon @40 |
| 7 | PMS2 | Frameshift | T3, N0, M0 | Sigmoid | 64 | Male | None |

[a]Other cancer reported for patient and the age of diagnosis.

**Table 2 Non-Lynch syndrome patient information and tumor characteristics.**

| Patient | Gene[a] | Mutation | Cancer stage | Cancer site | Age at diagnosis | Gender | Other cancer[b] |
|---|---|---|---|---|---|---|---|
| 1 | ND | Non-Lynch | T3, N1, M0 | Sigmoid | 67 | Male | None |
| 2 | ND | Non-Lynch | T4, N2, M1 | Rectosigmoid | 29 | Female | None |
| 3 | ND | Non-Lynch | T3, N0, M0 | Rectum | 82 | Male | None |
| 4 | ND | Non-Lynch | T4, N2, M1 | Ascending | 62 | Female | Rectum @51 |
| 5 | ND | Non-Lynch | T3, N0, M0 | Sigmoid | 43 | Female | None |

ND not detected.
[a]No tumor genetic tests were performed on Non-Lynch syndrome patients.
[b]Other cancer reported and the age of diagnosis.

cells, we tested whether antioxidants could affect CRC in Lynch syndrome models. We had previously tested the effects of L-NIL on CRC induction in MMR-mutant mice, and found no effect[26]. Hence, we focused our efforts on the antioxidants VitC and NAC. We first tested whether VitC and NAC, administered for 3 weeks after weaning, could affect polyp numbers in $Apc^{\min/+} Msh2^{+/-}$ and $Apc^{\min/+} Msh2^{-/-}$ mice. We found that 3 weeks of treatment with either VitC or NAC had no impact on colonic and small intestinal polyp numbers in these mice (Fig. 7a), despite reducing 8-oxoG levels (Fig. 7e, Supplementary Fig. 9a). As we observed that VitC resulted in a small but insignificant decrease in small intestinal polyps in $Apc^{\min/+} Msh2^{+/-}$ mice, we also treated $Apc^{\min/+} Msh2^{+/-}$ mice with VitC for 12 weeks but again did not observe a significant difference in colonic or small intestinal polyp numbers (Fig. 7b). VitC also had not impact on polyp numbers in $Apc^{\min/+} Mlh1^{+/-}$ Lynch models (Fig. 7c), or in $Apc^{\min/+}$ mice alone (Supplementary Fig. 9b). We also tested whether VitC treatment affected polyp numbers in $Apc^{\min/+} Msh2^{\text{flox}/+}$ villin CRE mice, where only one allele of $Msh2$ is deleted in intestinal epithelial cells. This model gives approximately the same level of colonic polyposis as $Apc^{+/\min} Msh2^{+/-}$ mice (Supplementary Fig. 9c). We found that VitC reduced 8-oxoG levels (Supplementary Fig. 9d), and slightly reduced colonic polyps, but not small intestinal polyps (Fig. 7d). Collectively, these data show that antioxidants are effective at reducing oxidative DNA lesions, but do not consistently decrease colonic or small intestinal polyps in Lynch syndrome preclinical models, suggesting that oxidative DNA damage is not a critical mediator of CRC in these genetic backgrounds, as opposed to the colitis models.

## Discussion

ROS and RNI have been implicated as one of the mediators that facilitate cancer initiation during inflammatory processes. ROS and/or RNI can potentially potentiate tumorigenesis by affecting the activity of the ERK, Wnt, and Notch pathways[27,28], by transiently oxidizing thiol groups on sensor proteins that regulate MAPK and NF-κβ pathways[28], or by directly inducing DNA damage[4,17,29,30]. However, formal proof that the mutagenic effects of ROS and/or RNI lead to CAC and CRC is lacking. Our findings suggest that ROS and RNI produced at high levels during inflammation can generate genetic damage that precipitates CAC. We find, using three CAC models (infection with *Helicobacter* or *E. coli*/ETBF, or DSS treatment), that antioxidants or iNOS inhibitors reduce tumorigenesis and 8-oxoG levels in the colon without affecting other facets of inflammation, effectively separating these two phenomena. We cannot exclude other mechanisms of inflammation-induced tumor formation in the colon since

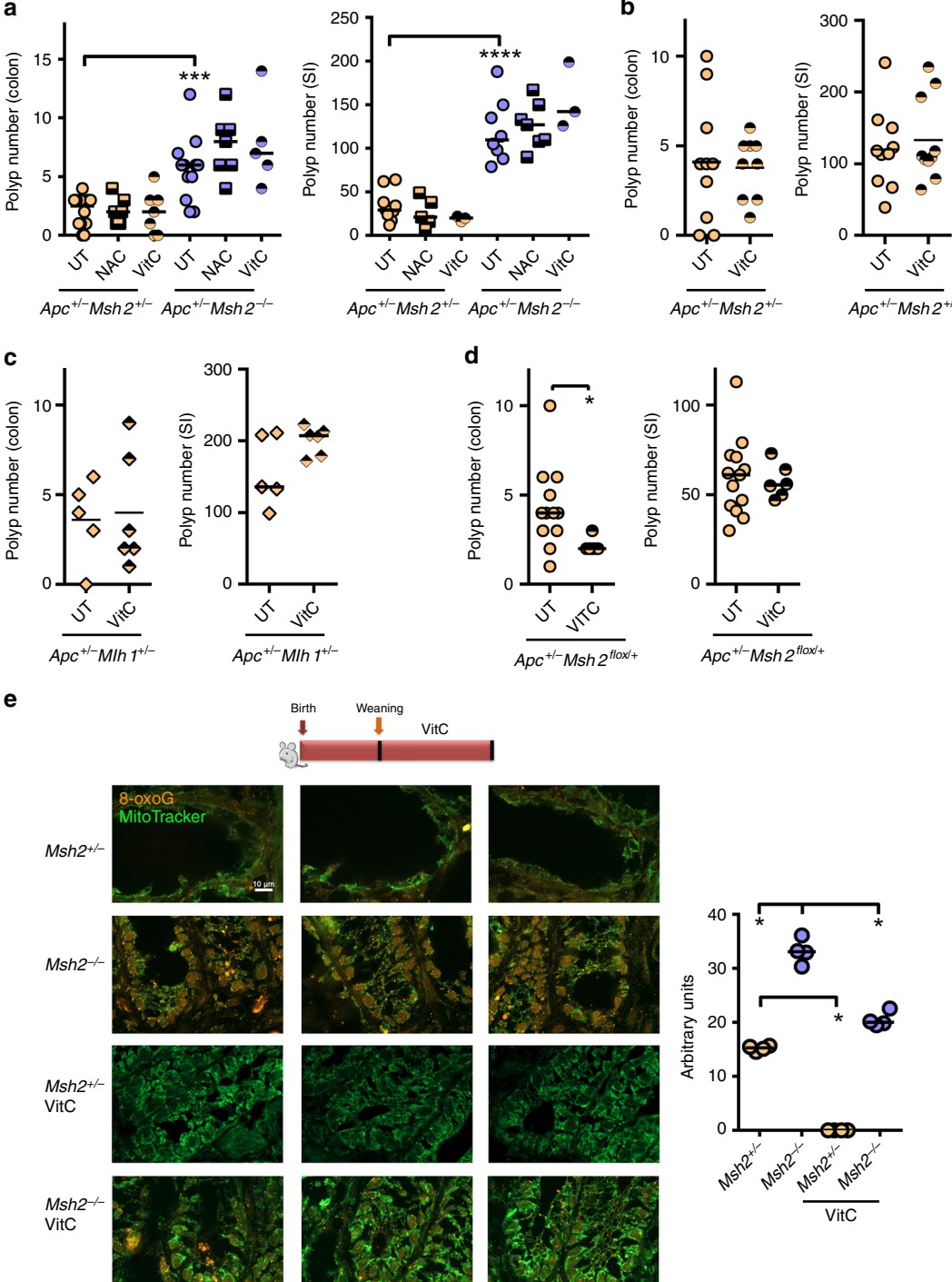

**Fig. 7 Antioxidants are not effective at reducing CRC in Lynch syndrome mouse models. a** Three-week-old $Apc^{min/+}$ $Msh2^{+/-}$ and $Apc^{min/+}$ $Msh2^{-/-}$ mice were treated with NAC or VitC for 3 weeks and polyp number was measured in the colon and small intestine (6-week-old mice). $N = 50$ mice were examined. **b** Four-week-old $Apc^{min/+}$ $Msh2^{+/-}$ mice were treated with VitC for 12 weeks and polyp number was measured in the colon and small intestine. $N = 20$ mice were examined. **c** Four-week-old $Apc^{min/+}$ $Mlh1^{+/-}$ mice were treated with VitC for 8 weeks and polyp number was measured in the colon and small intestine. $N = 11$ mice were examined. **d** Four-week-old $Apc^{min/+}$ $Msh2^{flox/+}$ villin CRE mice were treated with VitC for 12 weeks and polyp number was measured in the colon and small intestine. Each symbol represents the numbers of polyps in one mouse. $N = 18$ mice were examined. **e** Left panels: immunofluorescence for 8-oxoG and MitoTracker in colon from mice of the indicated treatments and genotypes. Representative images of four independent experiments, involving one mouse per group are depicted. Magnification 100×. Scale bar = 10 μm. Right panels: quantification of 8-oxoG immunofluorescence shown in left panels. One dot represents the median intensity of fluorescence of 40 nuclei per mouse as shown in Supplementary Fig. 9a. All data were analyzed using the two-sided non-parametric t-test Mann–Whitney; *$p < 0.05$, ***$p < 0.001$, ****$p < 0.0001$.

colitic IL10$^{-/-}$ mice treated with antioxidants or L-NIL had higher polyp counts than non-colitic IL10$^{-/-}$ mice.

The use of L-NIL, which blocks $\cdot$NO production by iNOS (ref. [31]), and NAC or VitC, which reduce both $O_2^{\cdot-}$ and ONOO$^-$, allows us to dissect the specific contributions of each radical to DNA damage. It also gives us the ability to infer the sources of genetic damage during normal metabolism or inflammation. Indeed, we found that L-NIL reduced CRC and 8-oxoG to the same extent as NAC or VitC (Figs. 2, 4 and 5), which suggests that $\cdot$NO is a major source of DNA damage caused by inflammation. Notably, iNOS is strongly upregulated in response to infection and $\cdot$NO is rapidly diffusible across biological membranes, suggesting that $\cdot$NO produced by immune cells can react with endogenous ROS produced within colonic epithelial cells to trigger mutation.

Colibactin, the genotoxin produced by *E. coli* NC101, has recently been shown to alkylate DNA (ref. [32]). Why then does colibactin-induced tumorigenesis depend on oxidative DNA damage in IL10$^{-/-}$ mice? Our data support the model proposed by Wilson et al.[32], in which oxidation, mediated by ROS/RNI, is necessary for the initial reaction that leads to the formation of colibactin–DNA adducts. It is also possible that different types of DNA lesions (e.g., adducts and oxidative DNA damage) act to synergistically promote oncogenesis by producing different genetic alterations (e.g., chromosomal translocations and point mutations).

It is worth noting that we previously found no role for RNI in tumor formation in homozygous null deficient *Msh2* mice[26], which is consistent with the notion that the Lynch syndrome model of CRC does not require inflammation. Notably, although VitC suppresses 8-oxoG, it did not have an effect on CRC in the Lynch syndrome model. This phenomenon was irrespective of whether one copy of the *Msh2* gene was disrupted or both alleles are removed. We hypothesize that the effects on CRC of reducing oxidative DNA damage in Lynch syndrome models is being masked by the tenfold increase mutation rate in *Msh2*$^{-/-}$ colon epithelial cells compared to controls[21], which is largely due to an inability to repair replication errors[33]. Unrepaired 8-oxoG leads to C:G > A:T transversion mutations, which account for only ~10% of all mutations in *Msh2*$^{-/-}$ cells[33,34]. Hence, in MMR-deficient mice, replication errors and spontaneous cytidine deamination dominate over the effects of oxidative damage[33,34], which might explain why antioxidants do not reduce CRC in Lynch syndrome models. Another possibility is that polyposis in Apc$^{min/+}$ MMR-heterozygous mice is purely affected by Apc loss of heterozygosity, and therefore any effects of antioxidants could be masked by Apc loss independent of oxidative DNA damage.

Our work indicates that the MMR system is the major repair mechanism for oxidative DNA damage in the gastrointestinal tract. Theoretically, ROS- or RNI-generated 8-oxoG could also be repaired by either OGG1- or MUTYH-initiated BER (ref. [6]). Although we expect that OGG1 and MUTYH are functional in *Msh2*$^{-/-}$ mice, our work shows that the BER system in colonocytes does not compensate for deficiencies in MMR. MUTYH/OGG1-deficient mice show an age dependent accumulation of 8-oxoG in the lung and small intestine[6], while *Msh2*$^{-/-}$ mice have increased 8-oxoG in other tissues, suggesting that repair of 8-oxoG is mediated by different combinations of repair pathways in different tissues[6].

We noticed that 8-oxoG appears stronger and punctuate, and localizes within mitochondria in DSS-treated mice (Fig. 4e). We speculate that this could be related to the higher inflammation observed in DSS-treated mice compared to infected mice. Higher inflammation might have resulted in higher amount of 8-oxoG in the tissue, and mutations of mitochondrial DNA (Fig. 4e; 8-oxoG staining co-localizes with MitoTracker FM).

We observed that although cytokine mRNA expression in DSS-treated IL10$^{-/-}$ mice was not significantly different than in DSS-treated IL10$^{-/-}$ mice administered with antioxidants, some cytokines were not significantly different in DSS-untreated mice compared with DSS-treated IL10$^{-/-}$ mice administered with antioxidants. While we do not know the reason for this result, one possibility is that antioxidants might have some anti-inflammatory effects in chemical-induced colitis models rather than in infection-induced colitis models.

In summary, DNA damage in colonocytes can be caused by ROS/RNI produced by inflammatory cells and endogenous ROS produced during the catabolism of microbial metabolites like butyrate. This oxidative/nitrosative DNA damage, which likely occurs throughout life in the colon, is constantly repaired by MMR. Although, our results suggest that both colitis and MMR-deficient models are susceptible to accumulate oxidative genetic lesions, antioxidants or molecules that inhibit iNOS reduce polyposis only in colitis models, suggesting that IBD patients might benefit from the use of such compounds. Future studies assessing the therapeutic value of antioxidants in CRC should be careful to stratify their study groups with respect to whether the underlying risk factors and genetic changes are due to oxidative stress.

## Methods

**Mice and treatments**. All mice were on the C57BL/6 J background. Apc$^{+/-}$ was used to denote Apc$^{Min/+}$ mice, which carry a nonsense mutation in exon 15 of the mouse *Apc* gene. Apc$^{Min/+}$ mice were purchased on The Jackson Laboratories. IL10$^{-/-}$ mice and *Msh2*$^{+/-}$ mice were provided by Dr. Ken Croitoru and Dr. Tak Wah Mak from University of Toronto, respectively. *Msh2*$^{+/-}$ villin CRE and *Mlh1*$^{-/-}$ mice were provided by Dr. Winfried Edelmann. Germ-free mice were purchased from McMaster University. IL10$^{+/-}$ uninfected were generated by crossing IL10$^{-/-}$ mice to germ-free mice. IL10$^{-/-}$ infected with *H. typhlonius* and *H. mastomyrinus* mice (IL10$^{-/-}$ I) were obtained by crossing IL10$^{-/-}$ with mice infected with these two bacterial strains leading to an infected line that was maintained separately from IL10$^{-/-}$ mice. IL10$^{-/-}$ I mice developed colitis spontaneously without any treatment. For DSS-induced colitis model, IL10$^{-/-}$ or IL10$^{+/-}$ mice were treated with 1% (w/v) DSS (molecular weight ranges from 36–50 kDa; MP BIOMEDICALS) in the drinking water for 1 week.

To measure colitis, each of the following parameters was given a value of either 0 or 1: stool with excreted mucus, rectal inflammation, rectal prolapse, bloody stool, over 5% weight lost, and moribund, leading to a maximum grade of 6. For rectal administration of butyrate (Sigma), 3-week-old mice were treated with 1 g/L ampicillin, 1 g/L metronidazole, and 1 g/L neomycin in their drinking water. Water was replaced twice a week until the end of the experiment. Six-week-old antibiotic-treated mice were administered with 100 μL rectal enemas containing 0.5 mM or PBS for three consecutives days. For antioxidant treatment, 3-week-old mice were administered 0.5% NAC, 330 mg/L VitC, or 500 μg/mL L-NIL (Cayman Chemical) in their drinking water. Freshly prepared antioxidants solutions were filtered and replaced twice a week until the end of the experiment. All mice, except IL10$^{-/-}$, were raised under specific pathogen-free conditions and fed a Teklad Global 18% protein rodent chow (Harlan, WI, USA). IL10$^{-/-}$ mice were raised under conventional conditions. All experimental animal procedures were approved by University of Toronto—University Animal Care Committee.

**Bacterial infection models**. *H. hepaticus* strain 3B1 (ATCC 51449) was obtained from ATCC. *H. hepaticus* was grown on brucella agar supplemented with 5% defibrinated sheep's blood at 37 ºC in a microaerobic environment (85% $N_2$, 10% $CO_2$, and 5% $O_2$). *H. hepaticus* was harvested after 4 days of growth and resuspended in PBS. Mice were inoculated with a bacterial suspension with an optical density of 1.0 at 600 nm (~$10^8$ CFU) in a volume of 0.2 mL. *E. coli* NC101 was provided by Dr. Christian Jobin (University of Florida, USA) and ETBF was provided by Dr. Cynthia L. Sears (Johns Hopkins University, USA). *E. coli* NC101 was grown overnight in LB broth under shaking conditions, while ETBF was grown in BHI broth (with Hemin, Vitamin K1 and Clindamycin) anaerobically at 37 ˚C (ref. [18]). Four-week-old IL10$^{-/-}$ mice were administered with water containing 500 mg/L cefoxitin for 48 h and then switched to water at least for 24 h before orally inoculating them with a mixture containing $5 \times 10^8$ cfu of each bacterium in 0.2 mL of PBS. To monitor the infection status throughout the experiment, fecal pellets were collected once a week and the presence or absence of *H. hepaticus*, *C. rodentium*, *E. coli* NC101, and ETBF were assessed by PCR (Supplementary Table 1), qPCR (Supplementary Table 2), or 16S rRNA gene sequencing on total gDNA, isolated from the stool samples (Genomic DNA from soil Kit, Macherey-Nagel).

**Quantitative real-time PCR to analyze relative abundance of Helicobacter**. Total DNA was extracted from the pellet samples using the Genomic DNA from soil Kit (Macherey-Nagel) and following manufacturer's instructions. A total of 20 ng/μL bacterial DNA was analyzed by qPCR using 16 S rDNA primers (Supplementary Table 2) to target *Helicobacter* groups[35]. Relative abundance of targeted bacterial groups and bacterial density were calculated according to previous studies[36]. Briefly, relative abundance was calculated by normalizing ΔCt for each target group to the Eubacteria (housekeeper) group and bacterial density was calculated from the total number of 16 S rRNA gene copies (eubacteria) based on each sample's DNA concentration and weight.

**Quantitative real-time PCR to analyze inflammatory signaling**. Total RNA was extracted from proximal part of colonic tissues using TRIzol® (Life Technologies) following the manufacturer's protocol. A total of 1 μg of DNA-free RNA was used for cDNA synthesis using Maxima H Minus reverse transcriptase (Thermo Fisher). For qPCR, gene-specific mRNA transcripts were amplified from cDNA in CFX384 Touch™ Real-Time PCR Detection System (BioRad) using SYBR FAST qPCR master mix (Kapa Biosystems) and specific primers (Supplementary Table 3). The specificity of PCR products was verified by melting curve analysis. Relative quantitation was done by the comparative CT method. Data presented as fold change after normalization with HPRT expression.

**16 S rRNA gene analysis of bacteria**. The V4 hypervariable region of the 16 S rRNA gene was amplified using a universal forward sequencing primer and a uniquely barcoded reverse sequencing primer to allow for multiplexing[37]. Amplification reactions were performed using 12.5 μL of KAPA2G Robust HotStart ReadyMix (KAPA Biosystems), 1.5 μL of 10 μM forward and reverse primers, 7.5 μL of sterile water, and 2 μL of DNA. The V4 region was amplified by cycling the reaction at 95 °C for 3 min, 24× cycles of 95 °C for 15 s, 50 °C for 15 s, and 72 °C for 15 s, followed by a 5 min 72 °C extension. All amplification reactions were done in triplicate, checked on a 1% agarose TBE gel, and then pooled to reduce amplification bias. Pooled triplicates were quantified using Quant-it PicoGreen dsDNA Assay (Thermo Fisher Scientific) and combined by even concentrations. The final library was purified using Ampure XP beads (Agencourt), selecting for the bacterial V4 amplified band. The purified library was quantified using Qubit dsDNA Assay (Thermo Fisher Scientific) and loaded on to the Illumina MiSeq for sequencing, according to manufacturer instructions (Illumina, San Diego, CA). Sequencing was performed using the V2 (150 bp × 2) chemistry.

The UNOISE pipeline, available through USEARCH version 9.2, was used for sequence analysis[38]. The last base, typically error-prone, was removed from all the sequences. Sequences were assembled, and quality trimmed using –fastq_mergepairs and –fastq_filter, with a –fastq_maxee set at 1.0 and 0.5, respectively. Assembled sequences <233 bp were removed. Following the UNOISE pipeline, merged pairs were then dereplicated and sorted to remove singletons using the vsearch software[39]. Sequences were denoised and chimeras were removed using the unoise2 command in USEARCH v. 9.2. Assembled sequences were then mapped back to the chimera-free denoised sequences at 97% identity OTUs, using vsearch. Taxonomy assignment was executed using utax, available through USEARCH, and the UNOISE compatible Ribosomal Database Project database version 16, with a minimum confidence cutoff of 0.9 (ref. [40]). OTU sequences were aligned using PyNast accessed through QIIME (ref. [41]). Sequences that did not align were removed from the dataset and a phylogenetic tree of the filtered aligned sequence data was made using FastTree[42]. Low abundance OTUs (<0.005% RA) were removed from the OTU table[43]. Alpha diversity, beta diversity and rarefactions were calculated using QIIME (ref. [41]). The data was rarefied to an even depth of 10,000 sequences per sample. Principal Coordinate Analysis plots were of the rarefied data using QIIME and plotted using EMPeror[44].

**ROS measurement**. Six-week-old mice were euthanized and dissected. Colons were inverted, placed on a 3 mm stirring rod and incubated with 30 mM EDTA at 37 °C for 30 min. Colons were then washed with PBS and the crypts were resuspended by aggressively shaking the colons in PBS for 1 min. Colonic crypts were washed and disrupted into single cells by incubation with 1X TRYPLE express (Invitrogen) for 10 min at 37 °C. Colonocytes were passed through a syringe, filtered through a 40 μM cell strainer, and resuspended in Phenol red-free Advanced F12 medium (Thermo Fisher) containing antibiotics without serum. Extracellular $H_2O_2$ production was measured using the Amplex Red Hydrogen Peroxide/Peroxidase Assay kit (Invitrogen). Briefly, colonocytes were treated with 0.1% NAC, 0.5 mM sodium butyrate, or PBS. Reactions were started by adding 20 μl of $1.5 \times 10^4$ colonocytes to 100 μl of reaction buffer. Fluorescence was measured at excitation/emission 530/590 nm during 30 min using a microplate reader (SpectraMax 300). Intracellular $H_2O_2$ production was measured using CM-$H_2$DCFDA (Invitrogen). Briefly, colonocytes were incubated with 5 μM CM-DCFDA during 30 min at 37 °C. Colonocytes were washed with medium and seeded at $2 \times 10^4$ per well. Cells were treated with either 0.1% NAC, 2 mM sodium butyrate, or PBS and fluorescence was measured at 495/520 nm.

**Immunofluorescence**. At 6 weeks of age or 24 h after butyrate treatment mice were euthanized, dissected, and colons were frozen in OCT (Optimal Cutting

Temperature Compound, Thermo Scientific). Swiss-roll-frozen colons were cut into 5 μM sections and placed on microslides. To stain mitochondria independently of their membrane potential, colon sections were incubated with 200 nM of MitoTracker FM (Molecular Probes, USA) for 15 min at room temperature (RT). MitoTracker solution was gently removed and the sections were fixed with 4% paraformaldehyde, washed with TBS- 0.1% tween, and permeabilized with 0.5% 100X Triton for 15 min. To detect oxidized DNA, sections were incubated with 100 μg/mL RNase A for 1 h at 37 °C and 10 μg/mL Proteinase K for 10 min[45]. DNA was unwound by a brief incubation with 2 M HCl for 5 min, followed by neutralization with 1 M Tris-base for 7 min. Unspecific binding was blocked by incubation with Goat F(ab) Anti-Mouse IgG H&L in 10% BSA (Abcam, 1:100) for 1 h at RT. Sections were incubated overnight with anti-8-oxoG antibody (QED Bioscience, 1:100) at RT, followed by incubation with a PE-tagged anti-mouse IgG2b antibody (BioLegend, 1:400). DNA was stained with DAPI and the slides were mounted with Fluoromount (Sigma). Fluorescent images were obtained with an AxioObserverZ1 spinning disk confocal microscope. Intensity of fluorescence of ten nuclei per field, and four field per mouse was analyzed using ImageJ. To detect double-strand DNA breaks, colon sections were incubated overnight with anti-γH2AX Ser139 antibody (EMD Millipore, 1:100) at 4 °C, followed by incubation with an Alexa488-tagged anti-mouse IgG1 antibody (Thermo Fisher Scientific, 1:400).

Fresh-frozen human CRC samples were obtained from the LTRI-Biospecimen Repository and Processing Lab, and were processed for 8-oxoG detection.

**Dot blot assay**. To validate anti-8-oxoG antibody specificity, five oligonucleotides: ATCGx5, ATCCx5, AAAAx5, TTTTx5, and CCCCx5 were treated with $H_2O_2$ and blotted in serial dilutions on a nitrocellulose membrane. The membrane was exposed to UV light for 5 min to crosslink the oligonucleotides, and blocked by incubation with 10% BSA for 1 h at RT. After washing with PBS, the membrane was incubated with anti-8-oxoG antibody (1:1000) for 1 h at RT, followed by incubation with an HRP-tagged goat anti-mouse antibody (Southern Biotech, 1:4000). The membrane was incubated with Clarity Western ECL (Biorad) and luminescence was detected with a Microchemi 4.2 Bioimaging system (DNR).

**Ex vivo oxidative phosphorylation assay**. Mice were treated with an antibiotic cocktail (see mice and treatments section). At 6 weeks of age, mice were euthanized and dissected. Crypts were extracted by incubating colons with 30 mM EDTA (37 °C for 30 min) in presence of: 0.5 mM butyrate, 100 μM palmitate, or 2 μM TSA (trichostatin A). After incubation crypts were extracted by shaking the colons in PBS. Then, crypts were washed and incubated in dark with a mix containing 200 nM Mitotracker Red CMXRos (Molecular Probes, USA), which stains metabolically active mitochondria, and 100 nM Mitotracker Deep Red FM (Molecular Probes, USA), which stains all mitochondria. After 15 min incubation, crypts were washed, cytospined, fixed, and counterstained with DAPI for analysis by immunofluorescence. Intensity of fluorescence of CMXRos in 15 cells per field, and three fields per treatment were analyzed using ImageJ.

**Histopathology**. The intestines were flushed with cold PBS, opened and polyps were measured. Then, the colons were prepared by the Swiss roll method, embedded in OCT, and rapidly frozen. 5 μm sections were cut on a cryostat and stained with hematoxylin and eosin. Pathologic evaluation of acute and chronic inflammation was performed by microscopy in a blinded manner. Acute inflammation was solely based on infiltration of neutrophils, which are very rare in healthy tissue. Chronic inflammation was based on infiltration of plasma cells and lymphocytes. Past the proximal colon there are very few plasma cells in the deep lamina propria. So chronic inflammation was defined by the presence of plasma cells (and lymphocytes) past the proximal colon in the bottom 1/3 of the lamina propria of the mucosa. Acute and chronic inflammation were independently semiquantitatively scored from 0–3 (0 = no significant inflammation, 1 = focal rare inflammatory cells, 2 = moderate numbers of inflammatory cells, and 3 = severe inflammation).

**Statistical analysis**. Data were analyzed using GraphPad Prism version 7.0. Unless otherwise indicated, all data were analyzed using the non-parametric *t*-tests (Mann–Whitney); *$p < 0.05$, **$p < 0.01$, ***$p < 0.001$, ****$p < 0.0001$. All error bars represent SEM.

**Reporting summary**. Further information on research design is available in the Nature Research Reporting Summary linked to this article.

## Data availability

The data that support the findings of this study are available from the corresponding author upon reasonable request. The authors declare that the data supporting the findings of this study are available within the paper and its supplementary information files.

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

## Acknowledgements

We are thankful to Drs. J. Carlyle, J. Danska, J. Brumell, P. Poussier, K. Croitoru, D. Philpott, and the Martin lab for their helpful comments. We thank Dr. Cynthia Sears and Dr. Christian Jobin for providing us the bacterial strains ETBF and *E. coli* NC101, respectively. We thank Cary Greenberg for her assistance with human tissue biospecimens. The human research reported in this publication was supported by the National Cancer Institute of the National Institutes of Health under Award Number U01CA167551 and through a cooperative agreement with the Ontario Familial Colorectal Cancer Registry (NCI/NIH U01/U24 CA074783). Additional support for case ascertainment was provided from the Ontario Cancer Registry, Canada. The content of this manuscript does not necessarily reflect the views or policies of the National Cancer Institute or any of the collaborating centers in the Colon Cancer Family Registry (CCFR), nor does mention of trade names, commercial products, or organizations imply endorsement by the US Government, the Ontario Cancer Registry, or the CCFR. The remaining work was funded by the Canadian Cancer Society (grant 703185) and Canadian Institute of Health Research (grant 144628).

## Author contributions

T. I., B.K.T., and M.K. designed the experiments, performed the research, and analyzed the data; J.C. performed the research and analyzed data; Y.M., C.S., and E.O.Y.W. performed the research; R.G. D.S.G., W.W.N., and A.M. supervised; and T.I. and A.M. wrote the manuscript.

## Competing interests

The authors declare no competing interests.

**Additional information**

