## [Peer Review File · Nature Communications]

Reviewers' comments:

Reviewer #1 (Expertise: Microbiome, inflammation/cancer immunology, Remarks to the Author):

The manuscript of Irrazabal et al investigates the mechanisms leading to CAC development in susceptible hosts, how inflammation and microbiota-derived metabolic products contribute to the process and the role of DNA damage in intestinal tumors formation.

The scientific question is of interest and the various in vivo models are relevant to address it and convincingly demonstrate the efficacy of antioxidant compounds in inhibiting tumor progression.

Nevertheless, some clarifications are required to fully convey the conclusions proposed by the authors. For instance, three main points require to be better addressed:

i) In the first part of the manuscript the authors evaluate, in different murine models, both the inflammation and the development of tumors. Colon length, though, is only a macroscopic indicator of overt inflammation. In the experiments involving antioxidants, authors should evaluate also different read-outs of colonic inflammation, i.e. mucosal expression of pro-inflammatory molecules (by RNA and/or protein) and immune cells infiltration.

Similarly, in the E.coli/B.fragilis infection model the caecum weight as inflammation evidence should be complemented with other more specific read-outs of inflammation.

ii) Authors do not provide data indicating that the butyrate is the causative metabolite linked to polyps formation in their in vivo models. They should provide evidences that, in vivo, butyrate is inducing DNA damage and hence polyps formation.

iii) Patient's data are interesting but do not provide any clear connection with butyrate production within lesions. Authors should provide evidences that in patients a correlation exists between microbiota-derived ROS-inducing metabolites and DNA-damage.

Minor points:

In the experiments with ApcMin/Villin Cre mice, authors show that antioxidant treatment reduce the tumor burden, but do not provide evidences of a direct correlation with microbiota-derived metabolites. Authors should better clarify this point.

Reviewer #2 (Expertise: Inflammation and DNA damage, Remarks to the Author):

In the manuscript by Irrazabal et al entitled “Limiting oxidative DNA damage reduces microbe-induced colorectal cancer” the authors determine that antioxidants and an iNOS inhibitor reduce 8-oxoguanine levels and polyp formation in different models of colitis associated tumorigenesis. They also link oxidative DNA damage to tumorigenesis in the setting of mismatch repair deficiency. The development of colitis and polyps in the mouse models used have been demonstrated elsewhere and NAC and Vitamin have been used in DSS models to reduce colon inflammation and tumorigenesis. Novelty for this manuscript lies in tying in the connection to oxidative DNA damage (see concern below) to the inflammatory models and a Lynch syndrome model. The manuscript is well-written and utilizes several mouse models but some concerns exist as indicated below:

Major concerns:

- 1) All of the inflammatory models rely on Il10^{-/-} mice, which have an altered immune response to infection. It is possible that all findings in these mice would be different in mice (or humans) with normal Il10 expression.
- 2) All of the oxidative damage studies rely on 8-oxoG staining of tissue sections. 8-oxoG antibodies are not highly specific and all manipulations of tissue/cells will produce additional oxidative damage. Another assay needs to be done to verify these findings. Some of the images in the manuscript are inconsistent as to what the staining should look like. For example, Figure 2d has strong nuclear staining whereas in Figure 4d the staining is throughout the cells and more punctate. The determining of intensity of fluorescence in only 10 nuclei per field seems arbitrary. Plus, the graphs of this data are unclear. For example, in Figure 2C the figure legend states that the images are representative of 5 experiments with >4 mice each and each point on the graph is 40 nuclei and the methods state that per mouse 10 nuclei were counted per image with 4 images per mouse. So, there should be >20 data points that each represent 40 nuclei but the graph only has 4 data points per condition.
- 3) The scoring used to determine colitis is rather subjective and colon length is not a direct measure of inflammation. The cytokines assayed in Supplemental Figure 4b are a more quantitative and direct way to measure the inflammatory response. However, it is concerning that some of these cytokines do appear to be affected by the treatments. For example, IL-1B appears significantly lower in the L-Nil and VitC groups than the H2O group. All statistically significant changes need to be identified. Plus, a similar quantitative assay such as IL-1B, IL6, Tnfa expression should be added to the other models to assess inflammation. This addition is important because in other DSS colitis models Vitamin C and NAC have been shown to reduce inflammatory cytokine expression and the finding that the antioxidants and the iNos inhibitor do not alter inflammation is important to the conclusions of this manuscript.

4) The introduction could use more detail as to the evidence suggesting MMR over BER for the repair of oxidative lesions in cells/tissue. This would better support the statement on line 70 that DNA damage (caused by ROS/RNI) would be repaired by the MMR system.

5) Figure 1A: what is the N for the points in the plot? Is the error plotted?

6) In Supplemental Figure 1A some of the IL10^{-/-} samples are missing bands for the presence of the relevant bacteria. There are also faint bands in the IL10^{-/-} Helicobacter genus samples. Based on the information include in the methods as to how these infected mice are generated (through breeding infected mice) being sure that all are infected as expected is important (including those treated with the antioxidants).

7) Throughout all of the figures it is difficult to know which comparisons are significant and which are not. Often when there is a control present the significance between the control and all treatment conditions is not clear. Figure 4C for example.

8) As the researchers mention in the beginning of the discussion, ROS/RNI can induce tumorigenesis by many potential methods. They demonstrate that antioxidants reduce 8-oxoG levels and polyp formation, but that does not prove that "oxidative DNA damage is the primary mechanism for neoplasia in an abiotic model of CAC" Line 157. Reducing ROS levels could reduce both 8-oxoG and activation of signaling pathways and then reduce polyp formation for example.

9) The baseline tumor numbers for the IL10^{-/-} DSS model are very different between Figure 3e and 4a (mean of approx.4 vs 20).

10) For figure 7, Apc^{+/-} mice need to be included as a reference point for the Apc^{+/-}Msh2f/+ mice. Do the Apc^{+/-}Msh2f/+ have more tumors and/or oxidative damage than the Apc^{+/-}? Does Vitamin C treatment bring them back to baseline or somewhere in between?

Minor comments:

1) All images or panels of images need scale bars.

2) There are several different alleles of mutant Apc available. Please include which truncation is present in the mice being used in the method section.

3) For the left most graphs in Figure 2a and b, are the mice infected with the combination of the bacteria? If so, that needs to be made clearer.

Reviewer #3 (Expertise: Microbiome/colitis/immunology, Remarks to the Author):

In this paper, Irrazabal and colleagues aimed at the role of Reactive oxygen species in colorectal cancer

Using three different models of colitis-associated colon cancer, the authors showed that treatment with iNOS inhibitor L-N6-(1-Iminoethyl) lysine dihydrochloride (L-NIL) or the antioxidants N-acetyl cysteine (NAC) or VitC decrease the colorectal carcinogenesis process without affecting colonic inflammation. This effect was associated with a decreased ROS-induced DNA lesion as showed by a decreased staining of 8-oxoG in the nuclei of colon epithelial.

The authors then showed that butyrate, a microbiota derived short chain fatty acid, increase the endogenous production of ROS by intestinal epithelial cells via its mitochondrial oxidative metabolism. In Msh2^{-/-} colon epithelial cells, butyrate induces ROS can even lead to the accumulation of oxidative DNA damage. Finally, the authors showed some results suggesting a relevance in human with Lynch syndrome.

Major comments :

- The authors showed that IL10 KO mice that received DSS developed colonic polyps. Can the author show histology of these polyps? Are they really neoplastic lesions?

- The pro-carcinogen role of Butyrate pointed out by the authors is intriguing. Particularly in colitis-associated colon cancer, the role of butyrate should be discussed as its production is usually decreased in intestinal inflammation. Moreover, the fact that ROS and iNOS inhibitors have an effect on the carcinogenesis and not on inflammation is surprising as butyrate concentration is supposed to decrease with intestinal inflammation. Butyrate (and other short chain fatty acids) dosage should be performed in the different experiments to support the conclusions.

- The authors showed data from patients with Lynch syndrome. It would be important to get similar analysis in patients with sporadic colorectal cancer and colitis associated colon cancer

Minor comments :

- Page 6 : “composition of the fecal bacteria” should be “composition of the fecal bacterial microbiota”

- Page 7 : unclear: “at which point inflammation”

Preamble: We would like to thank the reviewer's comments since we sincerely believe that their combined efforts have substantially improved the quality of the manuscript. Two of the main findings reported in this paper is that oxidative DNA damage is responsible for both colitis-associated colon cancer, as well as colorectal cancer in mismatch repair (MMR) mutant mice. However, in doing additional experiments for this manuscript, our previous conclusion that oxidative DNA damage is causing colon cancer in MMR mutant models is no longer supported by the data. That is, while Vitamin C-treatment led to a modest reduction in polyp counts in *Apc^{min/+} Msh2^{+/fl}* Villin-cre mice, our new data shows that Vitamin C or N-acetyl Cysteine treatment has no effect on polyps in *Apc^{min/+} Msh2^{-/-}* mice, *Apc^{min/+} Msh2^{+/-}* mice or in *Apc^{min/+} Mlh1^{+/-}* mice. This is despite our findings that the gut microbiota stimulates oxidative DNA damage in MMR-mutant mice, and that the MMR pathway is critical for repairing oxidative DNA damage in both mouse and human colonic tissue. As a result, we have now tempered our conclusion related to the relevance of oxidative DNA damage in Lynch (MMR-mutant) models, and thereby concluded that antioxidants or inhibitors of iNOS can prevent the development of cancer in colitis models of colon cancer, but not Lynch syndrome (mismatch repair mutant) models of colon cancer.

In addition to these experiments described above, we have carried out other experiments requested by the reviewers. We have appended the comments made by the reviewers below from the August 2019 review. The text changes in the manuscript that are a result of the reviewers' suggestions have been highlighted in **red**.

Reviewer #1:

The scientific question is of interest and the various in vivo models are relevant to address it and convincingly demonstrate the efficacy of antioxidant compounds in inhibiting tumor progression. Nevertheless, some clarifications are required to fully convey the conclusions proposed by the authors. For instance, three main points require to be better addressed:

i) In the first part of the manuscript the authors evaluate, in different murine models, both the inflammation and the development of tumors. Colon length, though, is only a macroscopic indicator of overt inflammation. In the experiments involving antioxidants, authors should evaluate also different read-outs of colonic inflammation, i.e. mucosal expression of pro-inflammatory molecules (by RNA and/or protein) and immune cells infiltration. Similarly, in the E.coli/B.fragilis infection model the caecum weight as inflammation evidence should be complemented with other more specific read-outs of inflammation.

Response: We thank the reviewer for this concern. We did look at other facets of inflammation, but these were "buried" in the supplemental figures. That is, in addition to colon length and colitis scores reported in **Figures 1 and 2** for the *Helicobacter*-infected mice, lymphocyte and neutrophil infiltration of colon tissue was blindly assessed by a pathologist (moved from

supplementary figures to **Figure 2b**). For the E.coli/B.fragilis model, in addition to the colon length, caecum weight, and blinded pathology report assessing lymphocyte and neutrophil recruitment, we also assessed the mRNA levels of inflammatory cytokines (moved from supplementary figures to **Figure 5a**). As such, we have addressed the reviewers concerns in two ways. First, we have moved these critical data from the supplementary section to the main figures. Second, for the DSS-treated mice and *Helicobacter*-infected mice, we have assessed the levels of the inflammatory cytokines (IL-6, TNF α , IL-12, IL1b, IL-17, and IFN γ) in colon tissue of mice untreated, or treated with antioxidants (**Figure 4a and Supplementary Figure 2a,b, respectively**) . All assays suggest that L-NIL or antioxidant treatment have no major effects on inflammation.

ii) Authors do not provide data indicating that the butyrate is the causative metabolite linked to polyps formation in their in vivo models. They should provide evidences that, in vivo, butyrate is inducing DNA damage and hence polyps formation.

Response: In the previous manuscript, we provided data showing that butyrate administered as an enema is directly inducing the 8oxo-G DNA lesion *in vivo* in mismatch repair-mutant mice. Since butyrate enemas require the mice to be under anesthesia, to directly test if butyrate can induce polyps, we would need 21 consecutive days of anesthesia and enemas, which was not allowed by our animal facility. As such, we were careful not to conclude that butyrate was responsible for cancer progression in MMR-mutant mice in the discussion of the previous version of the manuscript.

“However, we cannot exclude other sources of microbial-induced oxidative DNA damage in cells, and in fact, 3 days of butyrate treatment was not able to induce the same levels of 8-oxoG in antibiotic-treated Msh2^{-/-} mice as those found in untreated Msh2^{-/-} mice”

“In addition, since cancer was not an endpoint in the butyrate enema experiments, we cannot conclude the microbial-produced butyrate drives neoplasia in Lynch syndrome mouse models.”

In addition, as stated in the prologue above, our new data does not support a link between oxidative DNA damage caused by microbes in MMR-mutant mice and colon cancer, and hence, we have removed any such suggestions linking these two in the manuscript.

However, we did investigate whether butyrate leads to other kinds of DNA damage such as double stranded DNA breaks, which can be detected with γ H2AX. In a new **Supplementary Fig. 7b**, we show that γ H2AX is almost undetectable in colon epithelial cells of antibiotic- treated Msh2^{-/-} mice, and that physiological concentrations of butyrate induce γ H2AX foci formation.

iii) Patient's data are interesting but do not provide any clear connection with butyrate production within lesions. Authors should provide evidences that in patients a correlation exists between microbiota-derived ROS-inducing metabolites and DNA-damage.

Response: We have now included further data showing that tumors from non-Lynch patients do not have increased oxidative DNA damage, as we see in Lynch patient tumors, and in MMR-mutant mice (**Supplementary Fig. 8b**). This data further supports the notion that the MMR-pathway is critical in repairing oxidative DNA damage in the colonic tissue. However, as stated above, we now conclude that oxidative DNA damage does not promote colon cancer in MMR-mutant models, and as such, we now suggest that oxidative DNA damage is likely not relevant to cancer in Lynch patients.

Minor points:

In the experiments with ApcMin/Villin Cre mice, authors show that antioxidant treatment reduce the tumor burden, but do not provide evidences of a direct correlation with microbiota-derived metabolites. Authors should better clarify this point.

Response: This is a good point, but no longer relevant since we have removed this conclusion from the manuscript, as indicated above.

Reviewer #2:

Novelty for this manuscript lies in tying in the connection to oxidative DNA damage (see concern below) to the inflammatory models and a Lynch syndrome model. The manuscript is well-written and utilizes several mouse models but some concerns exist as indicated below:

1) All of the inflammatory models rely on IL10^{-/-} mice, which have an altered immune response to infection. It is possible that all findings in these mice would be different in mice (or humans) with normal IL10 expression.

Response: Thank you for these comments. Regarding the comment that the infections or DSS-treatment would only cause inflammation and polyps in IL10^{-/-} mice, this was precisely our point. IL10^{+/-} mice infected with *Helicobacter* or treated with DSS do not develop inflammation or polyps (**Figures 1 and 3**), while IL10^{-/-} mice do not develop polyps when they were not infected or not treated with DSS. Hence, these data support the notion that an inflammatory trigger (i.e. bacterial infection or DSS) in a genetically susceptible host promotes both sustained inflammation and cancer. In fact, bacterial infections with *Helicobacter*, *E.coli* and ETBF are not rare in the general population. What our results suggest is the interactions between specific pathobionts and host genetic mutations produce a "perfect storm" that results in inflammation and CAC.

However, not all pathobionts lead to this effect. We have now added new data showing that bacterial infection of IL10^{-/-} mice with *Citrobacter rodentium* does not lead to chronic inflammation nor to colon cancer (**Figure 1d, f**), supporting the idea that only specific pathogens combined with a specific genetic deficiency lead to colitis and colon cancer.

2) All of the oxidative damage studies rely on 8-oxoG staining of tissue sections. 8-oxoG antibodies are not highly specific and all manipulations of tissue/cells will produce additional oxidative damage. Another assay needs to be done to verify these findings. Some of the images in the manuscript are inconsistent as to what the staining should look like. For example, Figure 2d has strong nuclear staining whereas in Figure 4d the staining is throughout the cells and more punctate. The determining of intensity of fluorescence in only 10 nuclei per field seems arbitrary. Plus, the graphs of this data are unclear. For example, in Figure 2C the figure legend states that the images are representative of 5 experiments with >4 mice each and each point on the graph is 40 nuclei and the methods state that per mouse 10 nuclei were counted per image with 4 images per mouse. So, there should be >20 data points that each represent 40 nuclei but the graph only has 4 data points per condition.

Response: We thank the reviewer for this comment. Since our tissue samples are fixed before any manipulation is done, any 8-oxoG induced during the treatment would contribute to a background at an equal level in each sample. Therefore, differences in 8-oxoG levels cannot be attributed to sample manipulation as the background would be the same in each sample. In addition, we know that our antibody also recognizes 8-oxoG in mRNA, and when we treat the samples with RNase we can see that 8-oxoG staining disappears from the cytoplasm. We have included pictures of samples below not treated with RNase to support 8-oxoG specificity (**Figure R1**).

However, we acknowledge that it is possible that the 8-oxoG antibody might interact with other antigens. However, we provide new data and existing data that argue against this possibility. First, the MMR system has previously been suggested to repair the 8-oxoG lesion, and we find that the 8-oxoG lesion is dramatically elevated in MMR-mutant mice compared to controls (**Fig. 6a**), as well as in tumor tissue from Lynch syndrome patients, but not from non-Lynch patients (**Fig. 6b, Supplementary Fig. 8**, new data). Second, to assess the specificity of the 8-oxoG antibody, we designed oligonucleotides containing a known ratio of guanine bases, and treated them with hydrogen peroxide, which would induce ROS. We performed dot blots with this DNA and assessed whether the antibody is recognizing only oxidized guanine bases and whether hydrogen peroxide treatment correlates with a higher amount of 8-oxoG antibody detection. We found that oligonucleotides that do not have guanine are not recognized by the antibody, even if they are treated with hydrogen peroxide; and that hydrogen peroxide treatment of guanine-containing nucleotides increases the antibody binding (**Supplementary Figure 3a**). In addition, as mentioned before, we show that physiological concentrations of butyrate induce γH2AX foci formation, a double strand DNA break marker (**Supplementary**

Figure 7b), supporting the idea that butyrate can induce different types of DNA damage in *Msh2*-deficient colon epithelial cells.

We thank the reviewer for noticing that the figure legend was not clear. We have fixed this mistake by stating “Representative images of four independent experiments involving at least one mouse per group are depicted. Magnification 100X. Right panel: One dot represents the median intensity of fluorescence of 40 nuclei per mouse.” Figures legends 2, 4, 5, 6 and 7 have been altered in this way.

Figure R1. Representative images of 8-oxoG detection in DNA and RNA in colon of *Msh2*^{-/-} mice. 8-oxoG antibody can detect the antigen in both DNA and RNA (upper images). Therefore colon samples for all reported studies were first treated with RNase before DNA 8-oxo detection (lower images). Two mice per group. *Exposition time was adjusted to not saturate 8-oxoG intensity in RNase untreated samples.

3) The scoring use to determine colitis is rather subjective and colon length is not a direct measure of inflammation. The cytokines assayed in Supplemental Figure 4b are a more

quantitative and direct way to measure the inflammatory response. However, it is concerning that some of these cytokines do appear to be affected by the treatments. For example, IL-1B appears significantly lower in the L-Nil and VitC groups than the H2O group. All statistically significant changes need to be identified. Plus, a similar quantitative assay such as IL-1B, IL6, Tnfa expression should be added to the other models to assess inflammation. This addition is important because in other DSS colitis models Vitamin C and NAC have been shown to reduce inflammatory cytokine expression and the finding that the antioxidants and the iNos inhibitor do not alter inflammation is important to the conclusions of this manuscript.

Response: Thank you for this comment, and a similar point was raised by Reviewer 1 (point 1). As stated above, we have now assessed the mRNA levels of inflammatory cytokines in the colons of DSS-treated and Helicobacter-infected IL10^{-/-} mice, and we detect that antioxidants do not reduce these cytokines in the colon of these mice. Although in Supplemental Figure 4b (now **Figure 5a**), IL-1B appears to be lower, this is not statistically significant. IL-1B expression is regulated by several signalling pathways, some of which are activated by cytoplasmic DNA. Nuclear DNA damage can generate cytoplasmic DNA after chromosome mis-segregation and subsequent micronuclei rupture. Therefore, if antioxidants reduce nuclear DNA damage, they might indirectly reduce DNA sensor activation. However, our data does not support this conclusion in DSS-treated or pathogen-infected IL10^{-/-} mice. The difference is likely due to the chronic inflammation caused by these inflammatory triggers in IL10^{-/-} mice that are due to defective Treg activity that is not affected by antioxidants, and not inflammation caused by ROS-induced DNA damage, as might be the case for WT mice treated with DSS.

4) The introduction could use more detail as to the evidence suggesting MMR over BER for the repair of oxidative lesions in cells/tissue. This would better support the statement on line 70 that DNA damage (caused by ROS/RNI) would be repaired by the MMR system.

Response: Thank you for pointing this out. We have included this information in the introduction on paragraph 3.

5) Figure 1A: what is the N for the points in the plot? Is the error plotted?

Response: The N is 11 per group and for clarity the error was not plotted. We have now included the N number and error in the revised submission. 2-way ANOVA p<0.0001. Line 664.

6) In Supplemental Figure 1A some of the IL10^{-/-} samples are missing bands for the presence of the relevant bacteria. There are also faint bands in the IL10^{-/-} Helicobacter genus samples. Based on the information included in the methods as to how these infected mice are generated (through breeding infected mice) being sure that all are infected as expected is important (including those treated with the antioxidants).

Response: Thank you for this comment. IL10^{-/-} mice were infected with *H. mastomyrinus* and *H. typhlonius* by breeding with another mouse that was infected with both strains. We have also noticed that the abundance of *H. mastomyrinus* and *H. typhlonius* is variable in mice and as such, have been able to develop 3 cohorts of infected mice by breeding, those mice doubly infected with *H. mastomyrinus* and *H. typhlonius* (which we term IL10^{-/-}), and those singly infected with either of these bacteria, which were confirmed through this PCR assay. In addition, we have confirmed that infection is not affected by antioxidant treatment (**Supplementary Figure 2c**)

7) Throughout all of the figures it is difficult to know which comparisons are significant and which are not. Often when there is a control present the significance between the control and all treatment conditions is not clear. Figure 4C for example.

Response: In all our experiments, we have compared infected or DSS-treated mice vs uninfected or antioxidant/L-NIL treated mice. We see that it might be interesting to point out whether antioxidants revert a condition to untreated or uninfected levels. We avoided this comparison in the original manuscript as this can create cumbersome plots that statistically analyze all conditions to one another. Nevertheless, we have included other statistical comparisons of the data reported in the manuscript, such as in **Fig. 5a**, and new data such as in **Fig 4a**, and **Supplementary Fig. 2a, b**.

8) As the researchers mention in the beginning of the discussion, ROS/RNI can induce tumorigenesis by many potential methods. They demonstrate that antioxidants reduce 8-oxoG levels and polyp formation, but that does not prove that “oxidative DNA damage is the primary mechanism for neoplasia in an abiotic model of CAC” Line 157. Reducing ROS levels could reduce both 8-oxoG and activation of signaling pathways and then reduce polyp formation for example.

Response: While this is a formal possibility, the fact that both antioxidants and L-NIL, which have different modes of action, suppress polyp counts to a similar extent in all three colitis models (*Helicobacter*, DSS, and *E.coli*NC101/ETBF) suggest that reactive nitrogen intermediates (RNI) is responsible for cancer induction in these models (see 2nd paragraph in the discussion). We did use the word “suggest” in the discussion and have now tempered our conclusion on line 159.

9) The baseline tumor numbers for the IL10^{-/-} DSS model are very different between Figure 3e and 4a (mean of approx.4 vs 20).

Response: We are not sure why this is the case, and hence we did not want to speculate. While each experiment was internally controlled, the experiments detailed in **Fig. 3** and **Fig. 4** were performed ~2 years apart. One possible explanation for these differences in polyp number could be that we used two different lot numbers of DSS.

10) For figure 7, *Apc*^{+/-} mice need to be included as a reference point for the *Apc*^{+/-}-*Msh2*^{f/+} mice. Do the *Apc*^{+/-}-*Msh2*^{f/+} have more tumors and/or oxidative damage than the *Apc*^{+/-}? Does Vitamin C treatment bring them back to baseline or somewhere in between?

Response: *Apc*^{+/-}-*Msh2*^{f/+} have a slight increased number of polyps compared to *Apc*^{+/-} mice. Vitamin C does not have a significant effect in these mice. We have now included this data in **Supplementary Figure 9b**.

Minor comments:

1) All images or panels of images need scale bars.

Response: Thanks for noticing this, and we have now included these scale bars

2) There are several different alleles of mutant *Apc* available. Please include which truncation is present in the mice being used in the method section.

Response: We have included this information in the methods section. Line 328.

3) For the left most graphs in Figure 2a and b, are the mice infected with the combination of the bacteria? If so, that needs to be made clearer.

Response: Yes, and this has been fixed in the Figure legend.

Reviewer #3:

1. The authors showed that IL10 KO mice that received DSS developed colonic polyps. Can the author show histology of these polyps? Are they really neoplastic lesions?

Response: Thank you for this comment. We have now had our pathologist (Cathy Streukter) examine these tissues in a blinded manner, and include the results below in **Figure R2**. If the reviewer wishes that these data be included in the manuscript, we will do so. We have tried to minimize the amount of data added to the manuscript due to its current length.

IL10^{-/-} DSS dysplastic lesions

Figure R2. Representative images of dysplastic lesions in DSS-treated IL10^{-/-} mice. Dysplastic lesions were analyzed in a blinded manner and dysplasia was confirmed. Pictures depict dysplastic area found in three DSS-treated IL10^{-/-} mice.

2. The pro-carcinogen role of Butyrate pointed out by the authors is intriguing. Particularly in colitis-associated colon cancer, the role of butyrate should be discussed as its production is usually decreased in intestinal inflammation. Moreover, the fact that ROS and iNOS inhibitors have an effect on the carcinogenesis and not on inflammation is surprising as butyrate concentration is supposed to decrease with intestinal inflammation. Butyrate (and other short chain fatty acids) dosage should be performed in the different experiments to support the conclusions.

Response: We apologize for the confusion, but we did not suggest that butyrate causes oxidative DNA damage nor colon cancer in the IL10^{-/-} colitis models. As mentioned in the 2nd paragraph in the discussion, our data points to reactive nitrogen intermediates (RNI) as the major causative agents for the 8-oxoG lesion and polyps in the colitis models.

3. The authors showed data from patients with Lynch syndrome. It would be important to get similar analysis in patients with sporadic colorectal cancer and colitis associated colon cancer

Response: This is an excellent suggestion as we cannot exclude the possibility that the increased 8-oxoG lesion levels in tumors from Lynch patients is due to other factors that promote development of this lesion in tumor tissue that has nothing to do with the MMR-status of the tissue. Hence, we have now examined tumour tissue from non-Lynch patients and find that the levels of 8-oxoG are normal in these tissue (**Fig. 6b, Supplementary Fig. 8**). These results therefore support our findings that the MMR system is the major DNA repair pathway for repairing 8-oxoG damage in colon tissue, in both mice and humans.

Minor comments:

- Page 6 : “composition of the fecal bacteria” should be “composition of the fecal bacterial microbiota”
- Page 7 : unclear: “at which point inflammation”

Response: Thank you, these changes have been made.

Reviewers' comments:

Reviewer #1 (Remarks to the Author):

The authors replied to all my concerns in the revised version of the manuscript. I have nothing else to add.

Federica Facciotti

Reviewer #2 (Remarks to the Author):

In the revised manuscript by Irrazabal, some additional controls and experiments were provided to strengthen the rigor of the work. However, several points from the previous critique were not adequately addressed. There are still details and controls missing. Furthermore, the manuscript demonstrates correlation between oxidative damage and polyp formation but not causation. There are statements remaining that over-state their findings because of this lack of proof of causation.

1) Control groups are missing in many of the figures. Figure 1B IL10^{+/-} I (particularly needed because there is a trend to an increase in colitis score in 1A). Figures 1E IL10^{+/-} no infection (without this control it is not known whether or not infection alters colon length in the IL10^{+/-} background, which is necessary for the association between colitis and polyp formation – line 122). Figure 2a – uninfected negative control (it needs to be demonstrated that there was the expected decrease in colon length with infection in order to confident that there was no effect with NAC or N-NIL or VitC). Figure 2e – uninfected control. Figure 4b – uninfected control.

2) For figure 2, assays performed to determine changes in inflammation are not consistent between the different treatments used and it is not clear why.

3) For all of the bar graphs, the number of biological replicates used and the number of experiments they are from needs to be included. (For example, Supplementary Figures 1C, 2A, B, and Figures 4a, 5a).

4) As mentioned in the previous critique, for the cytokine expression data, it appears that the statistics were done by comparing each group to the infected H2O group. However, each group should also be compared to the untreated (uninfected) control as several of the cytokines do not appear to have a significant increase in expression in the antioxidant/infected group relative to the untreated (Supplementary Figure 2a, Figure 4a, 5a). Since a key point to the manuscript is that the antioxidant/infected groups still have inflammation, this is a necessary statistical comparison to include.

5) The 8-oxoG staining in figure 4E is very different than figure 2E (nuclear only). This was mentioned in the previous review but there was not an explanation provided.

6) In response to the previous critique data was provided for polyp number in Apc^{+/-} only mice (Supplementary Figure 9b). Since the polyp number is not different between these mice and Apc^{+/-}-Msh2^{+/-} mice (or Apc^{+/-}-Mlh1^{+/-}) it is not appropriate to imply that Apc^{+/-}-Msh2^{+/-} or Apc^{+/-}-Mlh1^{+/-} mice are a model of MMR deficiency. –

7) In the last sentence of the abstract, it is written that microbe-induced oxidative/nitrosative DNA damage play causative roles in inflammatory CRC models. While there is data presented correlating oxidative DNA damage and polyp formation, there is not a proof of causation in the work presented in this manuscript.

8) The statement in lines 94-96 is not supported by the data shown in this manuscript.

9) Lines 123-125 also state that inflammation and dysbiosis “leads to” the development of CAC. (It is also important to note that polyps formed in these models are tumors and not cancer).

Reviewer #3 (Remarks to the Author):

The authors adressed all my comments

Preamble: We would like to thank the reviewer 2 for their comments. We have addressed the comments below. The text changes in the manuscript that are a result of the reviewer's suggestions have been highlighted in red.

Reviewer #2 (Remarks to the Author):

In the revised manuscript by Irrazabal, some additional controls and experiments were provided to strengthen the rigor of the work. However, several points from the previous critique were not adequately addressed. There are still details and controls missing. Furthermore, the manuscript demonstrates correlation between oxidative damage and polyp formation but not causation. There are statements remaining that over-state their findings because of this lack of proof of causation.

1) Control groups are missing in many of the figures. Figure 1B IL10^{+/-} I (particularly needed because there is a trend to an increase in colitis score in 1A). Figures 1E IL10^{+/-} no infection (without this control it is not known whether or not infection alters colon length in the IL10^{+/-} background, which is necessary for the association between colitis and polyp formation – line 122). Figure 2a – uninfected negative control (it needs to be demonstrated that there was the expected decrease in colon length with infection in order to confident that there was no effect with NAC or N-NIL or VitC). Figure 2e – uninfected control. Figure 4b – uninfected control.

Although there is a trend towards an increase in colitis scores in *Helicobacter*-infected IL10^{+/-} mice, detectable signs of colitis started at around 15 weeks (see **Figure 1a**), and the reported colon length in **Figure 1b** and all subsequent experiments is data obtained from mice that were 9 to 10 weeks old, a point at which we did not detect visible signs of colitis in IL10^{+/-} mice.

In several independent experiments we showed that IL10^{-/-} uninfected mice have significantly longer colon than IL10^{-/-}-infected mice (**Figure 1b, e and 5b**). In Figure 2 mice were infected through breeding with a mix of *Helicobacter* species, and therefore there is no IL10^{-/-} uninfected littermate control mice available. As we included those controls when mice were infected through gavage (**Figures 1e, f, and 5b**), we believe that adding data from mice that are not littermate controls is not appropriate and does not bring any extra valuable information to the manuscript.

2) For figure 2, assays performed to determine changes in inflammation are not consistent between the different treatments used and it is not clear why.

We thank the reviewer for this comment and believe that the confusion is due to the 2 different groups of mice in this experiment that were carried out at different times. The first experiment included mice that were infected with a mix of *Helicobacter* species, and mice were divided in three groups: untreated, NAC or L-NIL. Our pathologist analyzed the tissue from this experiment in a blind manner, and we included tissue from IL10^{-/-} uninfected mice as an internal control. Using two measures of inflammation, including that of a pathologist, we found

little to no effect on inflammation in mice treated with L-NIL and NAC, but a strong effect on polyp number. In the second experiment, which was carried out after the first experiment, we were fortunate enough to have obtained mice that were infected with one or the other *Helicobacter* species through breeding. We took advantage of analyzing these mice in addition to the mice infected with both *Helicobacter* species. Since we already tested the effects of NAC or L-NIL treatment on inflammation and CRC in the mice coinfecting with both *Helicobacter* species, we decided to test whether VitC-treatment had an effect similar to NAC, and indeed, we found that this was the case. We have now clarified the reason for why these two groups of mice were treated differently in the results section on p. 9.

3) For all of the bar graphs, the number of biological replicates used and the number of experiments they are from needs to be included. (For example, Supplementary Figures 1C, 2A, B, and Figures 4a, 5a).

We thank the reviewer for noticing this omission. We have now included the following information in the figure legends: **Figure 1Sc** n per group ≥ 5 , **Figures S2a** and **S2b** n per group ≥ 4 , **Figure 4a** n per group =5, and **Figure 5a** n per group =6.

4) As mentioned in the previous critique, for the cytokine expression data, it appears that the statistics were done by comparing each group to the infected H₂O group. However, each group should also be compared to the untreated (uninfected) control as several of the cytokines do not appear to have a significant increase in expression in the antioxidant/infected group relative to the untreated (Supplementary Figure 2a, Figure 4a, 5a). Since a key point to the manuscript is that the antioxidant/infected groups still have inflammation, this is a necessary statistical comparison to include.

As stated in our previous rebuttal, we did not include comparisons between IL10^{-/-} untreated mice and IL10^{-/-} DSS-treated/infected plus antioxidants to not overcomplicate the Figures. However, we agree with the reviewer that this is an important point to show that the antioxidant-infected groups still have inflammation relative to uninfected controls. We now include these data in the manuscript (**Figures 4a, 5a, and Fig S2a,b**). We observe that antioxidants do not revert changes in cytokine expression induced after infection, however this effect is less pronounced in the DSS models (**Figure 4a**). That is, comparing antioxidant and DSS-treated mice to DSS-untreated mice results in the increased expression of a few inflammatory cytokines, not all of them as seen in the other models. This discussion has been added to the discussion section (2nd last paragraph).

5) The 8-oxoG staining in figure 4E is very different than figure 2E (nuclear only). This was mentioned in the previous review but there was not an explanation provided.

We do not have a verified explanation for this, however we believe that this might be due to the increased inflammation induced by DSS versus the infection models: cytokines expression

profiles (**Figure S2b vs Figure 4a**), colon length (**Figure 2a vs Figure 4b**), and inflammatory infiltrates (**Figure 2b vs Figure 4c**). We believe that higher inflammation in DSS-treated mice might have resulted in higher amount of 8-oxoG in the tissue, which causes mutations in mitochondrial DNA (**Figure 4e**). Indeed, the 8-oxoG staining co-localizes with mitochondria (MitoTracker FM in **Figure 4e**). We have included such a statement in the discussion section (3rd last paragraph), but this is still only speculation.

6) In response to the previous critique data was provided for polyp number in *Apc*^{+/-} only mice (Supplementary Figure 9b). Since the polyp number is not different between these mice and *Apc*^{+/-}*Msh2*^{+/-} mice (or *Apc*^{+/-}*Mlh1*^{+/-}) it is not appropriate to imply that *Apc*^{+/-}*Msh2*^{+/-} or *Apc*^{+/-}*Mlh1*^{+/-} mice are a model of MMR deficiency. –

We agree with the reviewer that it is not correct to suggest that *Apc*^{+/-}*Msh2*^{+/-} is a model for MMR-deficiency. However, we never stated this in the manuscript. We stated that such mice are models for Lynch syndrome.

7) In the last sentence of the abstract, it is written that microbe-induced oxidative/nitrosative DNA damage play causative roles in inflammatory CRC models. While there is data presented correlating oxidative DNA damage and polyp formation, there is not a proof of causation in the work presented in this manuscript.

We respectfully disagree with this reviewer's assessment. Some of our data is indeed correlative: in our 3 different CAC models, we find that oxidative DNA damage correlates with inflammation, and polyp counts. However, treating mice with 3 different agents each of which have different modes of action and all show the same effect on polyp count supports the conclusion being proposed in this manuscript. If we were to make this conclusion with only one such agent, then I would agree with the reviewer since it is possible that such an agent might have affected polyp count indirectly, and not through the proposed antioxidant effect or by inhibiting iNOS. However, the fact that these 3 different agents have similar effects makes it very difficult to come up with an alternative explanation for the observed effects.

8) The statement in lines 94-96 is not supported by the data shown in this manuscript.

Lines 94-96 states: "We show that gut microbiota, partially through the production of butyrate, induces ROS and the accumulation of 8-oxoG lesion and double strand DNA breaks in MMR-deficient cells."

In **Figure 6a** we show that treatment with antibiotics leads to a reduction in the oxidative DNA lesion 8-oxoG in the colon of *Msh2*^{-/-} mice. When these antibiotic-treated mice are given an enema with butyrate, 8-oxoG lesions reappear in the colon although not at the levels seen in untreated mice. In **Figure S6c** we show that treatment with butyrate or another fatty acid (palmitate), is sufficient to lead to ROS production in colon epithelial cells. In **Figure S7b** we

show that treatment with butyrate, lead to γ H2AX formation in colon epithelial cells, which is a known marker of double stranded DNA breaks. Therefore, we do not understand what part of this statement is not supported by our data as the data presented does support this statement.

9) Lines 123-125 also state that inflammation and dysbiosis “leads to” the development of CAC. (It is also important to note that polyps formed in these models are tumors and not cancer).

We agree and thank the reviewer for noticing this. We have changed “CAC” to “tumors” throughout the manuscript.

REVIEWERS' COMMENTS:

Reviewer #2 (Remarks to the Author):

My concerns were adequately addressed.

One thing for the authors to consider is that because there are basically the same number of colon polyps in the Apc-/+ mice as the Apc-/+Msh2+/- and Apc-/+Mlh1-/+ mice that all colon polyps formed in these mice are likely from loss of Apc and not from decreased levels of MSH2 or MLH1. Therefore, it is not surprising that the antioxidant treatment did not alter tumorigenesis. Granted the Apc-/+MSH2-/- mice do have more tumors than the Apc-/+ but this could be explained by an increase in Apc LOH that is independent from oxidative damage.

I would argue that to really call the Msh2 or Mlh1 heterozygous mice Lynch syndrome models there would need to be evidence that all tumors in these mice have inactivated the second Msh2 or Mlh1 allele (as is common in Lynch syndrome as stated by the authors).

REVIEWERS' COMMENTS:

Reviewer #2 (Remarks to the Author):

My concerns were adequately addressed.

One thing for the authors to consider is that because there are basically the same number of colon polyps in the Apc-/+ mice as the Apc-/+Msh2+/- and Apc-/+Mlh1-/+ mice that all colon polyps formed in these mice are likely from loss of Apc and not from decreased levels of MSH2 or MLH1. Therefore, it is not surprising that the antioxidant treatment did not alter tumorigenesis. Granted the Apc-/+MSH2-/- mice do have more tumors than the Apc-/+ but this could be explained by an increase in Apc LOH that is independent from oxidative damage.

I would argue that to really call the Msh2 or Mlh1 heterozygous mice Lynch syndrome models there would need to be evidence that all tumors in these mice have inactivated the second Msh2 or Mlh1 allele (as is common in Lynch syndrome as stated by the authors).

We thank the reviewer for this comment, and have addressed this issue in the discussion where we stated that: "Another possibility is that polyposis in Apc^{min/+} MMR-heterozygous mice is purely affected by Apc loss of heterozygosity, and therefore any effects of antioxidants could be masked by Apc loss independent of oxidative DNA damage."